# The N-terminus of varicella-zoster virus glycoprotein B has a functional role in fusion

Stefan L. Oliver[1]*, Yi Xing[2¤a], Dong-Hua Chen[3], Soung Hun Roh[4], Grigore D. Pintilie[5], David A. Bushnell[3], Marvin H. Sommer[1], Edward Yang[1], Andrea Carfi[2¤b], Wah Chiu[5,6,7‡], Ann M. Arvin[1,6‡]

1 Department of Pediatrics, Stanford University School of Medicine, Stanford, California, United States of America, 2 GSK Vaccines, Cambridge, Massachusetts, United States of America, 3 Structural Biology, Stanford University School of Medicine, Stanford, California, United States of America, 4 Department of Biological Sciences, Institute of Molecular Biology & Genetics, Seoul National University, Seoul, Korea, 5 Bioengineering, Stanford University School of Medicine, Stanford, California, United States of America, 6 Microbiology and Immunology, Stanford University School of Medicine, Stanford, California, United States of America, 7 Division of Cryo-EM and Bioimaging SSRL, SLAC National Accelerator Laboratory, Menlo Park, California, United States of America

¤a Current address: Syros Pharmaceuticals, Inc. Cambridge, Massachusetts, United States of America
¤b Current address: Moderna, Cambridge, Massachusetts, United States of America
‡ These authors jointly supervised this work.
* sloliver@stanford.edu

**Data Availability Statement:** X-ray crystallography data for the 2.4Å structure of VZV gB was deposited into the Protein Data Bank with accession code 6VLK. Cryo-EM maps and models

## Abstract

Varicella-zoster virus (VZV) is a medically important alphaherpesvirus that induces fusion of the virion envelope and the cell membrane during entry, and between cells to form polykaryocytes within infected tissues during pathogenesis. All members of the *Herpesviridae*, including VZV, have a conserved core fusion complex composed of glycoproteins, gB, gH and gL. The ectodomain of the primary fusogen, gB, has five domains, DI-V, of which DI contains the fusion loops needed for fusion function. We recently demonstrated that DIV is critical for fusion initiation, which was revealed by a 2.8Å structure of a VZV neutralizing mAb, 93k, bound to gB and mutagenesis of the gB-93k interface. To further assess the mechanism of mAb 93k neutralization, the binding site of a non-neutralizing mAb to gB, SG2, was compared to mAb 93k using single particle cryogenic electron microscopy (cryo-EM). The gB-SG2 interface partially overlapped with that of gB-93k but, unlike mAb 93k, mAb SG2 did not interact with the gB N-terminus, suggesting a potential role for the gB N-terminus in membrane fusion. The gB ectodomain structure in the absence of antibody was defined at near atomic resolution by single particle cryo-EM (3.9Å) of native, full-length gB purified from infected cells and by X-ray crystallography (2.4Å) of the transiently expressed ectodomain. Both structures revealed that the VZV gB N-terminus (aa72-114) was flexible based on the absence of visible structures in the cryo-EM or X-ray crystallography data but the presence of gB N-terminal peptides were confirmed by mass spectrometry. Notably, N-terminal residues $^{109}$KSQD$^{112}$ were predicted to form a small α-helix and alanine substitution of these residues abolished cell-cell fusion in a virus-free assay. Importantly, transferring the $^{109}$AAAA$^{112}$ mutation into the VZV genome significantly impaired viral propagation. These data establish a functional role for the gB N-terminus in membrane fusion broadly relevant to the *Herpesviridae*.

have been deposited in the Electron Microscopy Data Bank with accession codes 22519 (7.3Å gB-93k), 22520 (9.0Å gB-SG2) and 22629 (3.9Å structure of VZV gB; Protein Data Bank with accession code 7K1S). The authors declare that the data supporting the findings of this study are available within the paper and its Supplementary Information.

**Funding:** The National Institutes of Health (www. nih.gov) supported this research through grants P41-GM103832 (WC), R01-GM079429 (WC), R01-AI102546 (AMA), R37-AI20459 (AMA) and S10-OD021600 (WC). The funders had no role in study design, data collection and analysis, decision to publish, or preparation of the manuscript.

**Competing interests:** The authors have declared that no competing interests exist.

## Author summary

Herpesviruses are ubiquitous infectious agents of medical and economic importance, including varicella-zoster virus (VZV), which causes chicken pox and shingles. A unifying theme of herpesviruses is their mechanism of entry into host cells, membrane fusion, via a core complex of virally expressed envelope glycoproteins gB, gH and gL. Of these, the primary fusogen, gB, is activated by the heterodimer gH-gL through an unknown mechanism and enables the virus envelope to merge with cell membranes to release the DNA containing capsid into the cytoplasm to initiate infection. By using a human antibody that neutralizes VZV we have recently demonstrated that the initiation of membrane fusion is associated with the crown region of gB. Here, we use cryogenic electron microscopy to compare the structure of this human neutralizing antibody, 93k, to a non-neutralizing antibody SG2. Surprisingly, both antibodies bind to the crown of gB with considerable overlap of their footprints on gB with one important exception, SG2 does not bind to a flexible region in the gB N-terminus. Mutations incorporated into this flexible region disrupts gB mediated membrane fusion and significantly impairs VZV propagation, identifying an Achilles heel in viral replication.

## Introduction

Herpesviruses are enveloped double stranded DNA viruses that cause medically and economically important diseases in humans and animals [1]. Similar to other enveloped viruses, herpesviruses require virally encoded glycoproteins to penetrate cells via fusion between the virion and cell membranes to initiate and establish infection. A cooperative assembly of glycoproteins composed of the trimeric gB and the heterodimer gH-gL constitutes the core herpesvirus fusion complex. Due to structural similarities with viral fusogens of vesicular stomatitis virus (VSV) G protein and baculovirus gp64 [2–8], gB has been designated a type III fusogen. The gH-gL heterodimer is postulated to interact with host cell membrane proteins subsequently enabling membrane fusion by the gB trimer [9]. This interplay between gB and gH-gL along with the structural and dynamic properties of gB is poorly understood. However, based on studies of VSV G [2–7], gB has been proposed to transition from a prefusion to a postfusion state. During this transition, fusion loops within gB have been modelled to embed in the opposing cellular membrane so that the cell and virus membranes are brought into close proximity during a proposed gB condensation reaction where lipid mixing and pore formation occur [9].

Varicella-zoster virus (VZV) is a highly infectious, human host restricted alphaherpesvirus that causes varicella (chickenpox) upon primary infection, leading to subsequent infection of sensory ganglion neurons and establishment of latency where VZV can reactivate to manifest as zoster (shingles) [10]. As for other herpesviruses, VZV employs gB and gH-gL for virion entry fusion. Unlike the closely related herpes simplex virus (HSV) gB orthologues, the mature glycosylated form of VZV gB (~130kDa) is cleaved into two fragments (~69kDa and ~73kDa) at the $^{491}$RSRR$^{494}$ furin recognition site [11]. Importantly, the VZV gB/gH-gL fusion complex is required for cell-cell fusion (abbreviated as cell fusion) and the formation of polykaryocytes within varicella and zoster skin lesions, which is a hallmark of VZV pathogenesis and modeled *in vitro* by syncytia formation within infected MeWo cells [12,13]. Our previous studies have shown that VZV pathogenesis depends on the regulation of gB/gH-gL induced cell fusion via the carboxyl terminal domains of gB and gH [14–16]. In addition, the role of host factors in

VZV-induced cell fusion regulation are beginning to emerge as the loss of fusion regulation leads to altered gene transcription profiles, and virus-induced fusion is modulated by the host phosphatase, calcineurin [14,15]. Importantly, VZV overcomes the usual constraints against fusion between fully differentiated host cells leading to the potential for adverse health events. VZV reactivation has been associated with the debilitating chronic pain syndrome termed postherpetic neuralgia (PHN) likely as a result of fusion between ganglion neurons and satellite cells [16]. In addition, fusion between vascular endothelial cells leads to arteritis that has been linked to strokes [17–19].

Together with the VZV gH-gL heterodimer, VZV gB can trigger cell fusion *in vitro* in the absence of other viral proteins enabling fusion functions to be associated with the fusion complex directly [20–22]. We have recently established that gB domain IV (DIV), the crown, plays an essential role in fusion revealed by a 2.8Å cryo-EM structure of VZV gB in complex with the neutralizing human monoclonal antibody (mAb) 93k. This near atomic resolution structure and biochemical studies identified molecular contacts needed for antibody binding with gB DIV. Moreover, gB residues at the gB-93k interface were determined to be necessary for fusion function and VZV replication by mutagenesis of the VZV genome, indicating that DIV of herpesvirus gB plays an essential role in membrane fusion. Critically, the successful propagation of VZV *in vitro* and lesion formation in human skin xenografts *in vivo* is dependent on the regulation of cell fusion linked to the carboxyl terminal domain (CTD) of gB [23,24]. This was discovered through the use of hyperfusogenic gB CTD mutants that had multifactorial effects on the host cell transcriptome, were significantly impaired for replication *in vitro* and severely attenuated in human skin infection [23,24]. Similar regulatory roles have been proposed for HSV-1 and EBV gB CTDs and the membrane proximal region (MPR) of HSV-1 gB was shown to regulate association of the fusion loops with lipid membranes [25–32].

Unlike the human mAb 93k, an anti-VZV gB mouse mAb, SG2, did not inhibit fusion or prevent VZV infection of MeWo cells [33]. The purpose of the present study was to apply cryo-EM-based structural analyses to define the differential effects of these mAbs on inhibition of the gB fusogen. The SG2 mAb bound to gB DIV in close proximity to but did not fully overlap with the mAb 93k epitope. In contrast to mAb 93k, mAb SG2 did not bind to the VZV gB N-terminus, suggesting a potential role for this region in membrane fusion. Structures of native, full-length VZV gB (cryo-EM; 3.9Å) and the VZV gB ectodomain (X-ray crystallography; 2.4Å) were determined in the absence of bound antibodies to define the structure of the N-terminal region (amino acids (aa)72-114), which was found to be a part of DIV but flexible and not fully resolved by cryo-EM or X-ray crystallography. To investigate the relevance of the gB N-terminus in membrane fusion, site-directed mutagenesis was performed at a predicted α-helix formed by residues $^{109}$KSQD$^{112}$, which decreased or abolished cell fusion when gB was expressed with gH-gL in a virus-free assay, and significantly impaired the spread of VZV mutants in cultured cell monolayers. These findings corroborate the importance of gB DIV, providing further insight into a functional role of the N-terminus of herpesvirus gB in membrane fusion.

## Results

### Differential binding to VZV gB of neutralizing (93k) and non-neutralizing (SG2) mAbs at subnanometer resolution

Single particle cryo-EM maps were generated for native, full-length gB purified from infected MeWo cells in complex with Fab fragments of either the neutralizing mAb, 93k, or the non-neutralizing mAb, SG2, using the same electron microscope and data processing procedure (**Fig 1**, **S1 Table** and **S1** and **S2 Validation Reports**). Despite the failure of SG2 to neutralize

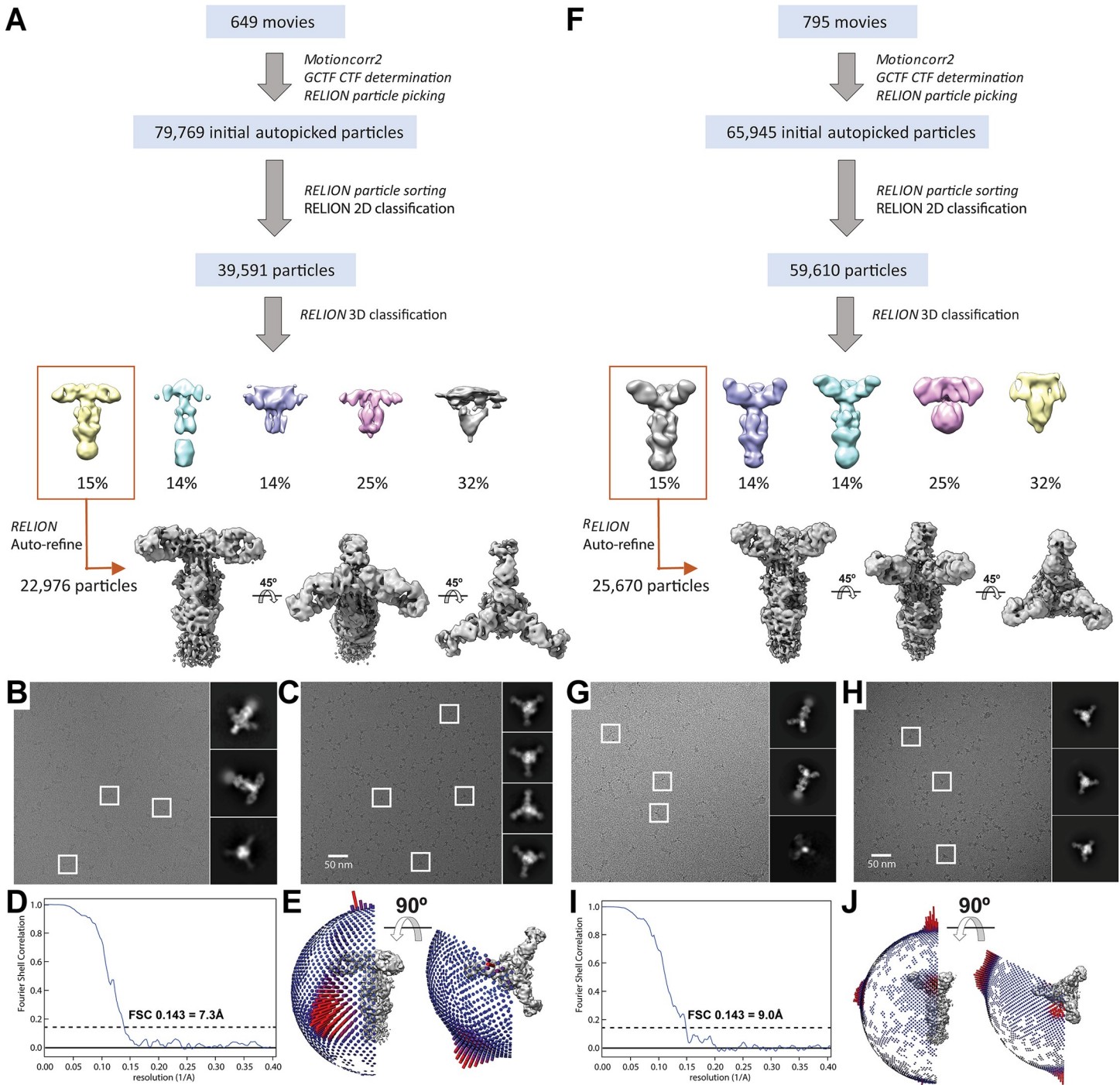

**Fig 1. Classification scheme of particles identified in cryo-EM micrographs and validation of subnanometer structures of either 93k (A to E) or SG2 (F to J) Fab fragments in complex with purified VZV gB.** B to J–Single particle cryo-EM micrographs of 93k (B and C) or SG2 (G and H) Fab fragments in complex with VZV gB in vitreous ice on lacey carbon (B and G) or Quantifoil (C and H) grids captured with a 200kV F20 (FEI). Representative 2D class averages are shown. Fourier shell correlation plot (D and I) and Euler angle distribution (E and J) of VZV gB-93k or gB-SG2 complex particles included in the 3D reconstruction.

VZV, the structures of gB-93k (7.3Å) and gB-SG2 (9.0Å) revealed that the SG2 epitope was also located in gB DIV and overlapped the 93k epitope (**Fig 2 and S1 Movie**). Though the two Fab complexes have different resolutions, the structures are resolved enough to visualize the

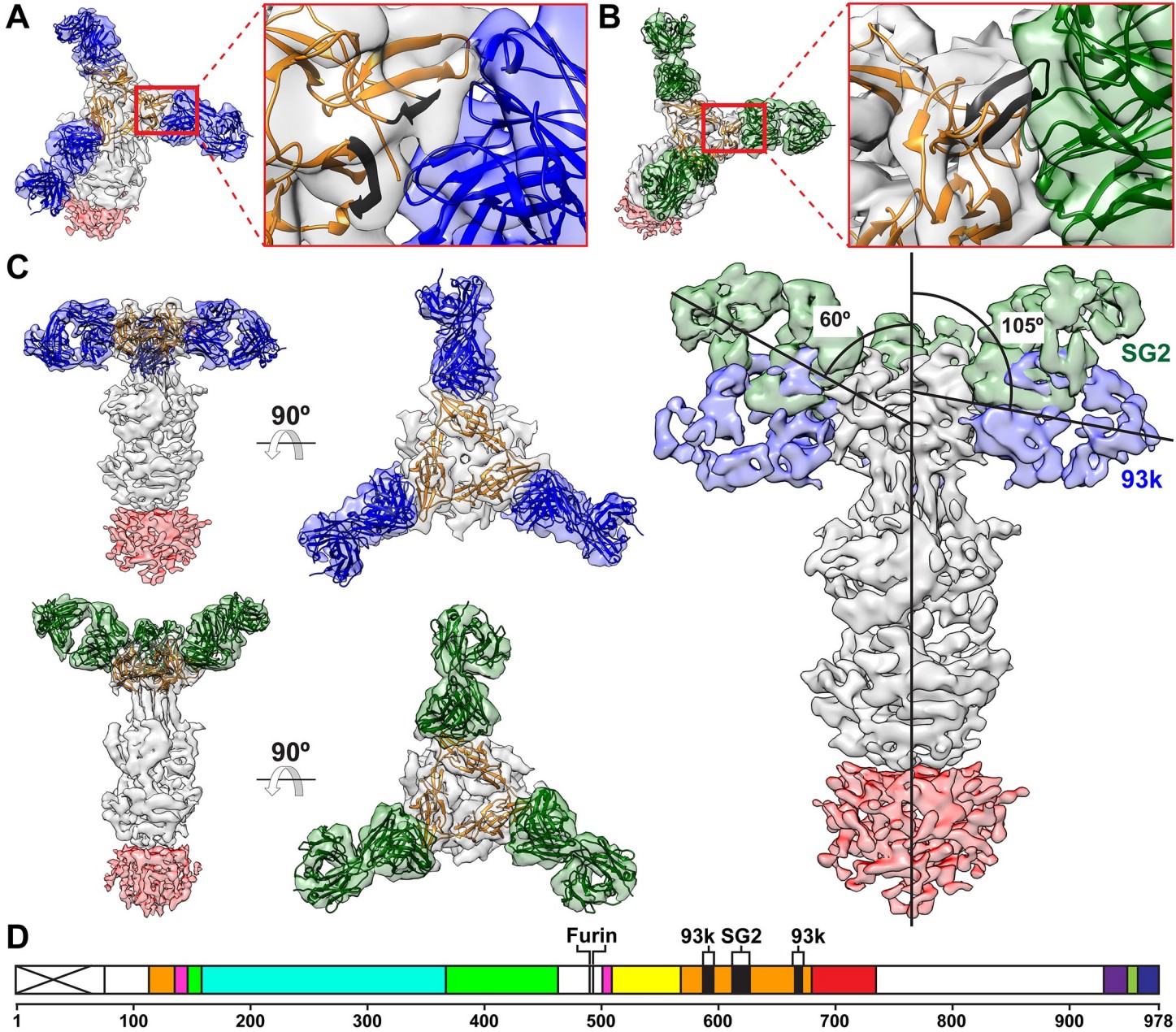

**Fig 2. Subnanometer resolution cryo-EM structures of Fab fragments from either 93k or SG2 in complex with native, full-length VZV gB purified from VZV infected MeWo cells.** A to C–Subnanometer resolution cryo-EM maps of gB-93k (93k –blue; 7.3Å) and gB-SG2 (SG2 –green; 9.0Å) complexes focused at the gB-Fab fragment interface (A and B), viewed from the side and top (C) and a composite cryo-EM map of the 93k and SG2 Fabs bound to native gB showing the difference in binding angles of the two Fabs. The Fab binding angles were calculated from the central axis of gB (vertical line) and the central axes of the 93k and SG2 Fabs. The gB ectodomain is shown in grey and the CTD show in red; the CTD is depicted at a different threshold to the ectodomain due to the lower resolution for this part of the cryo-EM map. D–A linear map of VZV gB showing the location of the mAb 93k major binding sites (β23 and β30) and the predicted mAb SG2 binding site (β25 and β26) in gB DIV. The gB furin cleavage site is shown. VZV gB domains are colored as follows, DI (cyan), DII (green), DIII (yellow), DIV (orange), DV (red) and linker regions (hot pink).

interactions between the Fab fragment and the gB protomer and reveal that the 93k Fab bound at an angle of 105° relative to the vertical axis of the gB trimer whereas the SG2 Fab bound at a 60° angle as shown in an integrated structure display in two different views (**Fig 2C**). As shown in our previous 2.8Å structure [33], the footprint of the 93k Fab encompassed the gB

N-terminal and C-terminal β-sheets, including β23, β25–26 and β28–30, and sealed the crevice between these β-sheets (**Fig 2**). In contrast, the footprint of the SG2 Fab primarily covered β25–26. These differences imply that the conformational change of the gB structure required for cell fusion is less dependent on β25–26 residues because their engagement by the SG2 mAb does not interfere with gB/gH-gL mediated fusion and support an essential role for the β23 and β30 residues in this critical process [33].

A second significant difference between the binding properties of mAbs 93k and SG2 found by comparing the gB-Fab structures was that the SG2 Fab does not make contact with the gB N-terminus (**Fig 3**). In our previous high-resolution structure of the gB-93k interface [33], the variable light chain cluster of determination region 2 (VLCDR2) of mAb 93k was shown to form interactions with gB T115 and K116. These residues are at the end of the N-terminal region of gB with a defined structure (**Fig 3A**). In support of our previous findings using the gB-93k map from the present study, the 93k-gB K116 interaction was also apparent (**Fig 3B**). In contrast, interactions of SG2 with the structured gB N-terminus were not identified at the gB-SG2 interface in the 9.0Å cryo-EM map (**Fig 3C**). Predicted contacts were only localized to the β24–26 region in DIV. Compared to mAb 93k, the absence of mAb SG2 binding at these gB N-terminal residues suggested that this region might contribute to the fusion function of gB.

## Characteristics of the VZV gB structure in the absence of bound antibodies

To determine whether 93k or SG2 Fab binding altered the conformation of the gB N-terminus, structures were determined in the absence of antibody by both single particle cryo-EM of native, full-length gB purified from infected cells (**Figs 4 and 5A–5D, S1 Table and S3 Validation Report**) and X-ray crystallography of the gB ectodomain produced by transient expression in HEK293T cells (**Fig 5E–5H, S2 Table and S4 Validation Report**). Full-length gB was visible as rod-like particles ~20 nm in length in vitreous ice on ultrathin carbon film supported on lacey carbon EM grids (**Fig 4C**). The cryo-EM structure of the native, full-length gB ectodomain was resolved at near atomic resolution (3.9Å) with an architecture resembling the postfusion form resolved for herpesvirus gB orthologues (**Fig 5B–5D**). Cryo-EM 2D class averages generated from 349,207 particles revealed four globular regions of gB, three in the ectodomain and one in the CTD (**Fig 4C**). The gB domains, DI-DV, were apparent in the cryo-EM map of the gB trimer constrained to a 3-fold (C3) axis of symmetry (**Fig 5A and 5C and S2 Movie**). In contrast to the ectodomain, the cryo-EM map resolution of the CTD, including the MPR, transmembrane (TM) and gBcyt, was >7.0Å due to extensive flexibility in the CTD that was apparent from the 2D class averages (**Figs 4C and 5B**).

The topologies of the gB ectodomain were similar for the structure derived from cryo-EM map (3.9Å) of the native, full-length gB and the X-ray crystallography structure (2.4Å) of the transiently expressed gB ectodomain (**Fig 5B–5D and 5F–5H and S2 and S3 Tables**). The gB cryo-EM structures did not differ in the presence or absence of 93k and SG2 Fabs, indicating that the gB ectodomain in its post fusion form was not altered by antibody binding. Two glycine substitutions in the gB fusion loop (W180G and Y185G) and three glycine substitutions at the furin cleavage site ($^{491}$RSRR$^{494}$ to $^{491}$GSGG$^{494}$) that were incorporated to improve protein solubility for X-ray crystallography did not affect the gB ectodomain when compared to the cryo-EM map of native, full-length gB (**S2 Movie**). The protomers of VZV gB assembled into trimers in the crystal as observed in the cryo-EM map and other herpesvirus gB orthologues [2–5,34].

The secondary structure of the gB ectodomain was well defined in the absence of gB antibodies both by cryo-EM and X-ray crystallography maps. These ectodomain structures were

## A - gB-93k (2.8Å)

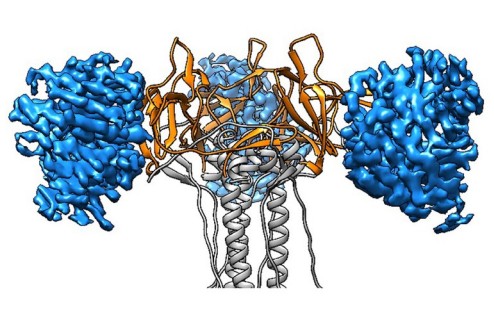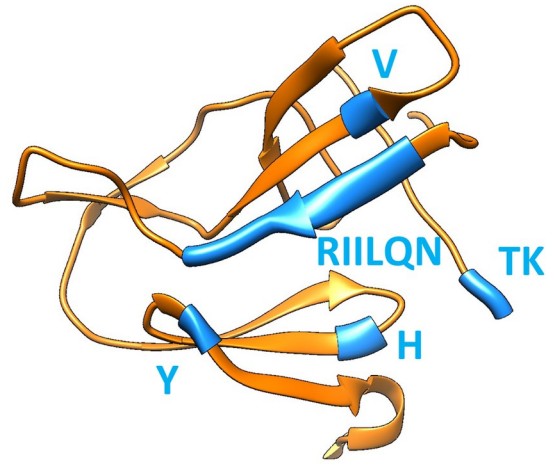

$^{115}$**TK**PT$^{118}$...$^{589}$SDT**RIILQN**SMRVS$^{602}$...$^{613}$LISI**V**SLNGSGT$^{624}$...$^{658}$**H**YVYYEDYR**Y**VREIA$^{672}$

## B - gB-93k (7.3Å)

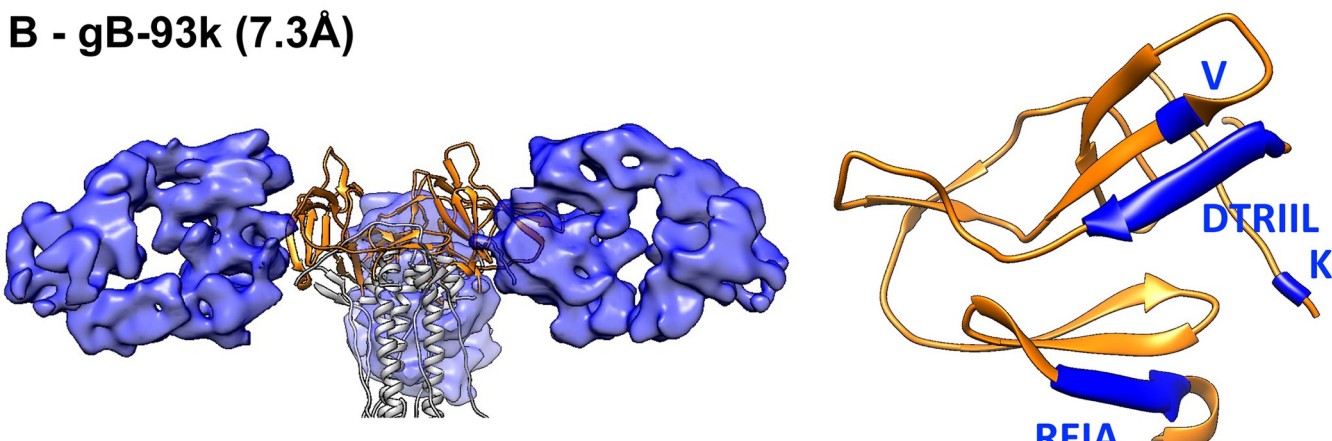

$^{115}$**T**K**PT$^{118}$...$^{589}$S**DTRIIL**QNSMRVS$^{602}$...$^{613}$LISI**V**SLNGSGT$^{624}$...$^{658}$HYVYYEDYRYV**REIA**$^{672}$

## C - gB-SG2 (9.0Å)

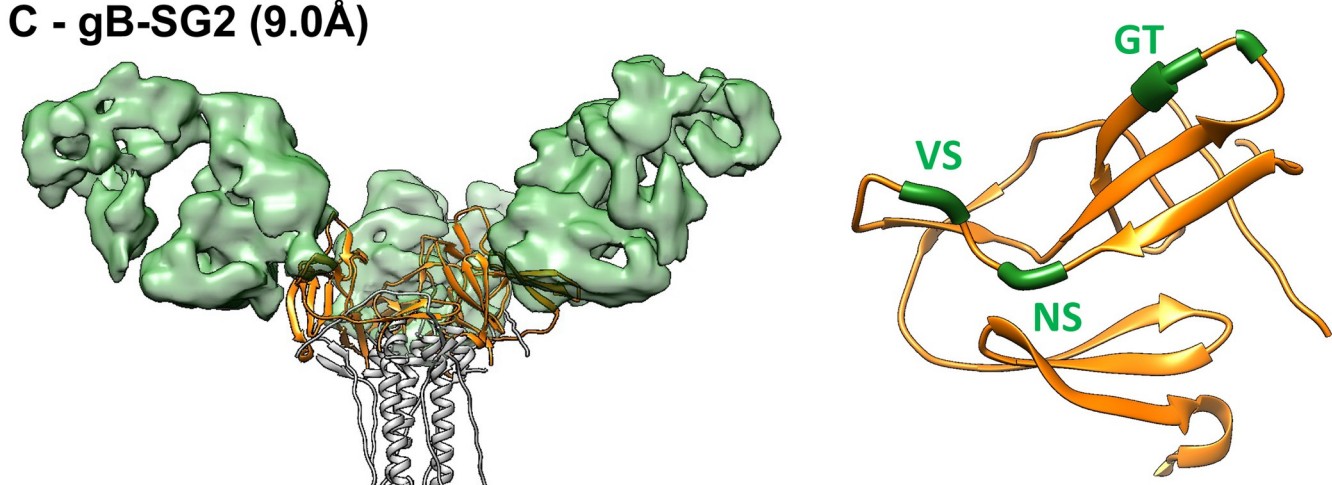

$^{115}$TKPT$^{118}$...$^{589}$SDTRIILQ**NS**MR**VS**$^{602}$...$^{613}$LISIVSLN**GS**G**T**$^{624}$...$^{658}$HYVYYEDYRYVREIA$^{672}$

**Fig 3. Anti-VZV gB monoclonal antibody interactions with gB DIV predicted from 93k-Fab and SG2-Fab subnanometer cryo-EM structures.** A to C– Cryo-EM maps of Fab fragments (93k at 2.8Å [33]–light blue; 93k at 7.3 Å–blue; SG2 at 9.0Å –green) bound to VZV gB DIV (orange). Interactions between Fab fragments from either 93k or SG2 with gB DIV domain (orange). The 93k and SG2 interactions are mapped on the gB DIV domain (orange ribbon), directly from segmented density of the Fab in maps of gB-93k 2.8Å (A–cyan), gB-93k 7.3Å (B–blue), and gB-SG2 9.0Å (C–green). Residues within 6Å of the segmented density in each respective case are colored on the ribbon and highlighted in the corresponding amino acid sequence below the cryo-EM and ribbon structures. Blue indicates residues that are predicted to interact with 93k, green indicates residues that are only predicted to interact with the SG2 Fab and not 93k (both 2.8Å and 7.3Å). Cryo-EM map thresholds: A– 0.03; B– 0.422; C– 2.52.

practically identical to the ectodomain of the high-resolution cryo-EM structure of gB-93k [33]. The five domains, DI-DV, were observed, including 13 helices and 31 β-strands (**Fig 5G, S2 Movie and S3 Table**). The core of DI in the crystal structure resembled a plekstrin homology (PH) domain fold, which is a putative phosphoinositide and protein binding site. The two proposed fusion loops, aa180-185 and aa264-269, were at the tips of two nearly orthogonal β-sheets in DI. The extended two-stranded β-sheet, aa336-343 in DI, remained intact despite the W180G and Y185G fusion loop substitutions. VZV gB DII, aa148-159 and 369–502, also formed a PH domain at its core. VZV gB is cleaved by furin at the $^{491}$RSRR$^{494}$ recognition site [11]. The aa455-502 region was disordered in the crystal structure despite the $^{491}$GSGG$^{494}$ sub-stitution of the furin cleavage site, which reduces gB cleavage and is dispensable for VZV

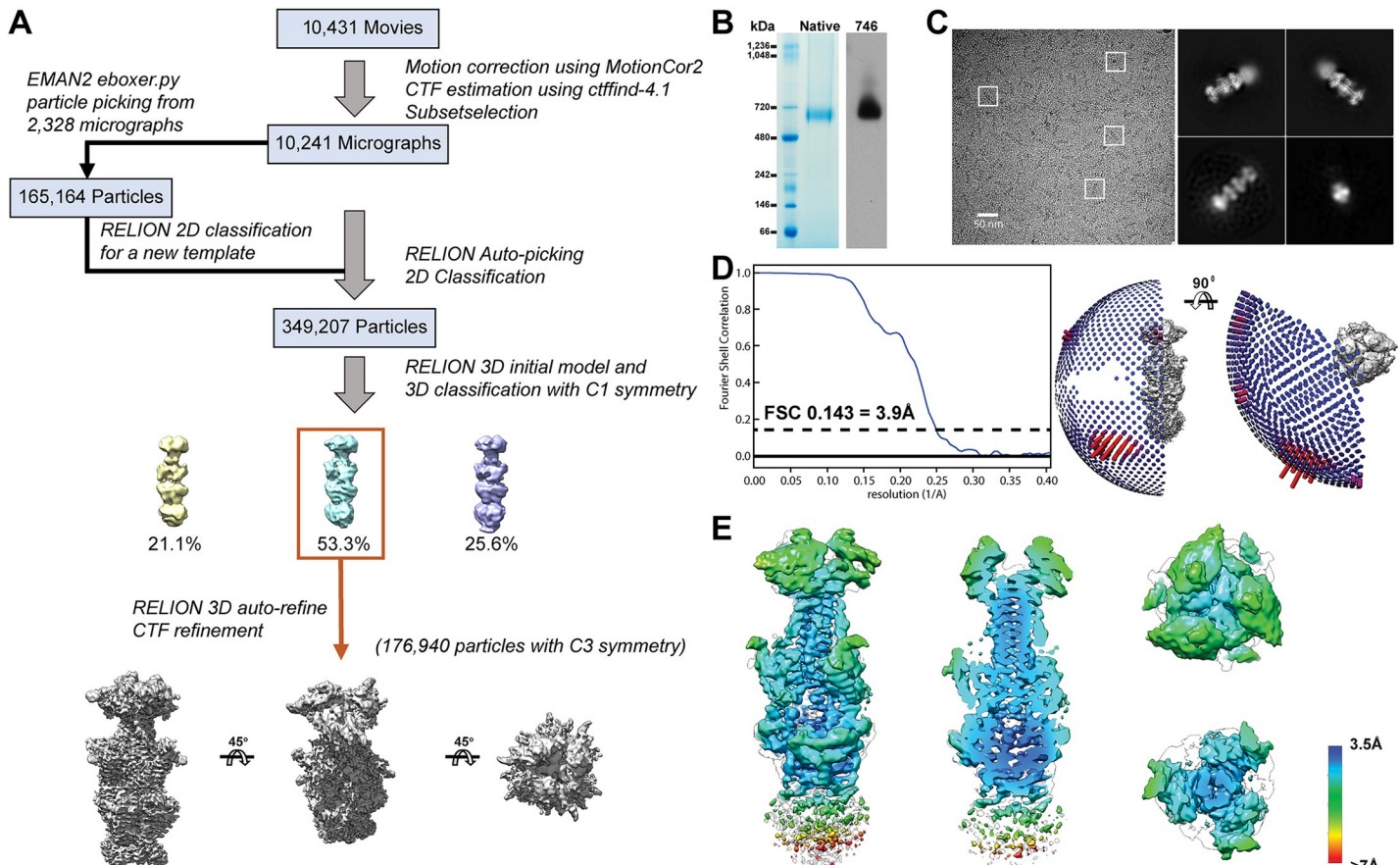

**Fig 4. Single particle cryo-EM of native, full-length VZV gB.** A–Classification scheme of VZV gB particles identified in Cryo-EM micrographs purified VZV gB frozen on lacey carbon grids. B–Native PAGE of purified VZV gB and western blots with VZV gB specific rabbit IgG 746–868. Molecular weights of the protein standards are given to the left of the gel (kDa). C–Single particle cryo-EM micrograph of a lacey carbon grid with purified VZV gB post size exclusion chromatography with four representative 2D class averages. D–Fourier shell correlation plot and Euler angle distribution of VZV gB particles included in the 3D reconstruction. E–Distribution of resolution (Å) of the VZV gB cryo-EM map calculated by ResMap.

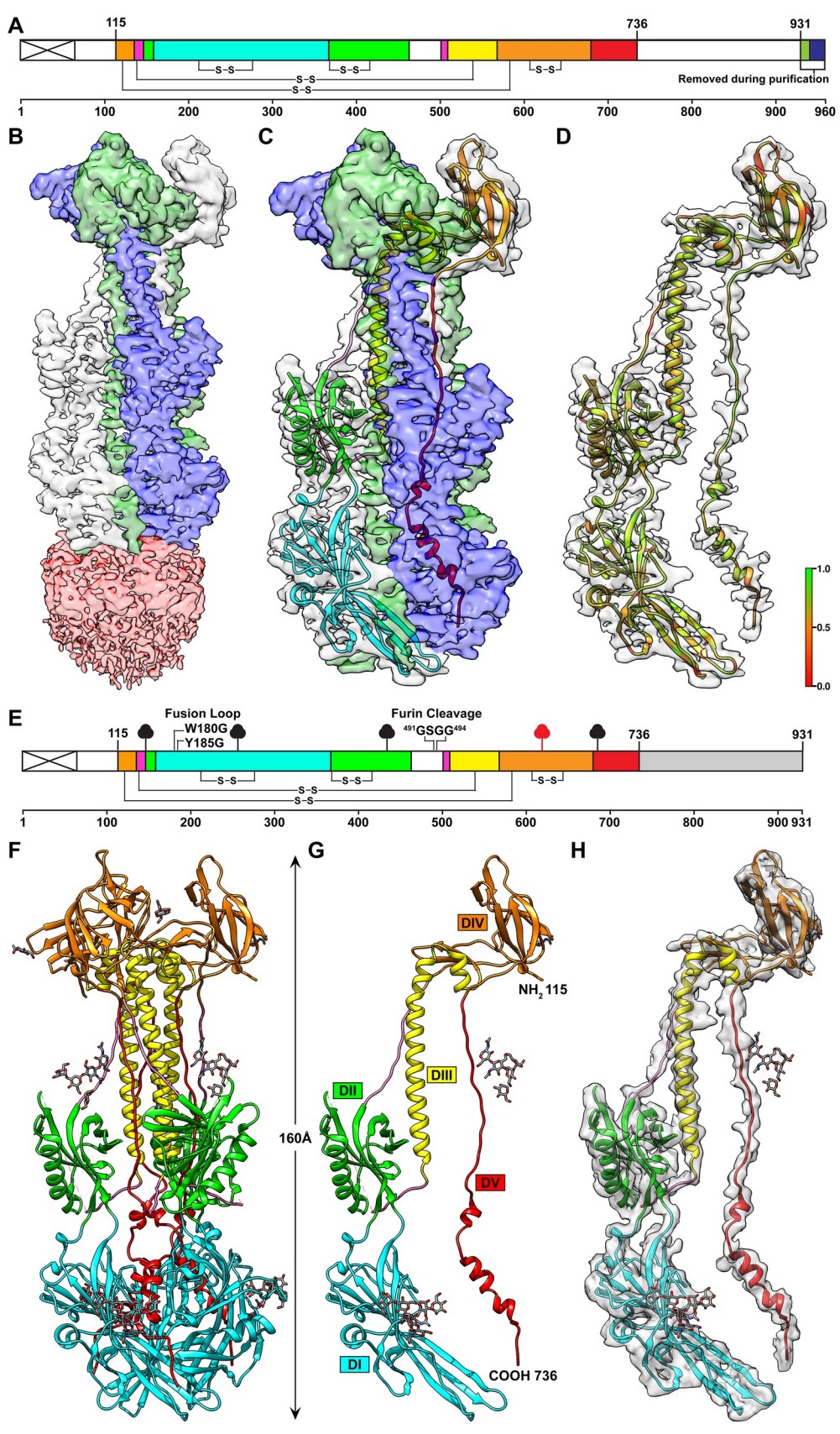

**Fig 5. Near atomic resolution structures of VZV gB native, full-length VZV gB purified from VZV infected MeWo cells and the ectodomain transiently expressed in HEK293 GnTI⁻ cells.** A–A diagram of the linear structure for VZV gB expressed by the pOka-gB-TEVV5 virus. The signal sequence is depicted by the crossed white box. Colored regions between residues 115–736 represent those that were resolved in the cryo-EM structure; DI (cyan), DII (green), DIII (yellow), DIV (orange), DV (red) and linker regions (hot pink). Disulphide bonds are represnted by the connecting lines. The colored regions beyond residue 931 represent the tag used for purification of gB from infected cells; TEV cleavage site (lime green) and V5 tag (blue). B–The cryo-EM map of native, full-length VZV gB constrained to C3 symmetry (3.9Å). The ectodomain is based on a focussed refinement map with each protomer of the gB ectodomain highlighted in different colors (blue, white and green) and the CTD represented at a lower threshold (pink). C–A ribbon diagram for the gB ectodomain structure is colored as for the diagram in A. D–Segmentaion and MapQ analysis of the VZV gB cryo-EM map based on a single protomer of the gB ectodomain. The scale, red to green (0 to 1), represents the goodness of fit of the cryo-EM map with the structure using MapQ. E–The linear structure of the truncated VZV gB used for X-ray crystallography. The signal sequence is depicted by the crossed white box. Colored regions between residues 115–736 represent those that were resolved in the gB crystal structure from transiently transfected cells; DI (cyan), DII (green), DIII (yellow), DIV (orange), DV (red) and linker regions (hot pink). The location of the residues that were substituted in the fusion loop (W180G and Y186G) and the furin cleavage site ($^{481}$GSGG$^{484}$) of the gB ectodomain expression construct used for X-ray crytsallography are indicated. The white regions represent portions of gB that were not resolved in the crystal structure. The grey shaded box indicates the truncation. The clubs represent glycosylation sites in the X-ray crystallography data (black) or by orbitrap mass spectrometry (red). F to H–The X-ray crystal structure of VZV gB ectodomain at 2.4Å. Ribbon diagrams of the gB trimer (F), monomer (G) and the monomer superimposed with a portion of a segmentation of the 3.9 Å cryo-EM map.

replication *in vitro* [11]. This site was also poorly resolved in the cryo-EM structure (**Fig 5B–5D** and **S2 Movie**), likely due to free amino and carboxyl termini resulting from furin cleavage of native gB. DIII helices, aa512-554 and 557–566, contributed to the central coiled-coil of the gB trimer. Fourteen β-strands of DIV with adjacent segments of DIII and DV formed the triangular crown region in the gB trimer. The extended loop of DV integrates into the groove of the coiled-coil of the trimer, and before the membrane proximal region ended at a C-terminal helix lying between DIs from two other protomers (**Fig 5F and 5G**).

Notably, the extreme N-terminus of gB was not resolved in either of the full-length or ectodomain only gB structures. The ectodomain crystal structure included aa115-736 but electron density was lacking for aa72-114 after the signal peptide, which was attributable to flexibility that was corroborated by the cryo-EM map (**Fig 5**). The Cα backbone could not be traced for aa72-114 in either the cryo-EM or X-ray crystallography datasets (**Fig 5**). To ensure that the extreme N-terminus was present, the natively expressed full-length gB purified from infected cells was digested with trypsin to generate peptides upstream of aa115 and detected using Orbitrap mass spectrometry (**Fig 6** and **S1 Spreadsheet**). Trypsin cleavage produced four peptides between aa94 and aa132, of which three were abundantly detected ($^{94}$SAHLGDG-DEIR$^{104}$, n = 131; $^{94}$SAHLGDGDEIREAIHK$^{109}$, n = 28; $^{105}$EAIHK$^{109}$, n = 1; $^{110}$SQDAETKPTFYVCPPPTGSTIVR$^{132}$; n = 163). Peptides were not detected for the portion of the gB N-terminus from aa27-93 due to an absence of trypsin cleavage sites (**Fig 6**). Importantly, the detection of peptides corresponding to aa94-114 verified the presence of the N-terminus in full-length gB and supports flexibility as the reason for failure to resolve this region in the gB structures.

## The VZV gB N-terminal region resembles herpesvirus orthologues

Overall, the VZV gB ectodomain structure closely resembled the X-ray crystal structures of PRV, HSV-1, EBV and HCMV gB orthologues, as demonstrated by sequence alignments and structural superpositions of the monomer (**Fig 7, S3 Movie** and **S5–S8 Tables**). Notably, the disordering of the region around the furin recognition site in VZV gB DII (aa455-502) was consistent with other herpesvirus gB orthologues, including HSV-1 which is not cleaved by furin, suggesting that this region has an intrinsic flexibility. Despite the W180G and Y185G substitutions, the X-ray crystallography structure of the VZV fusion loop was highly

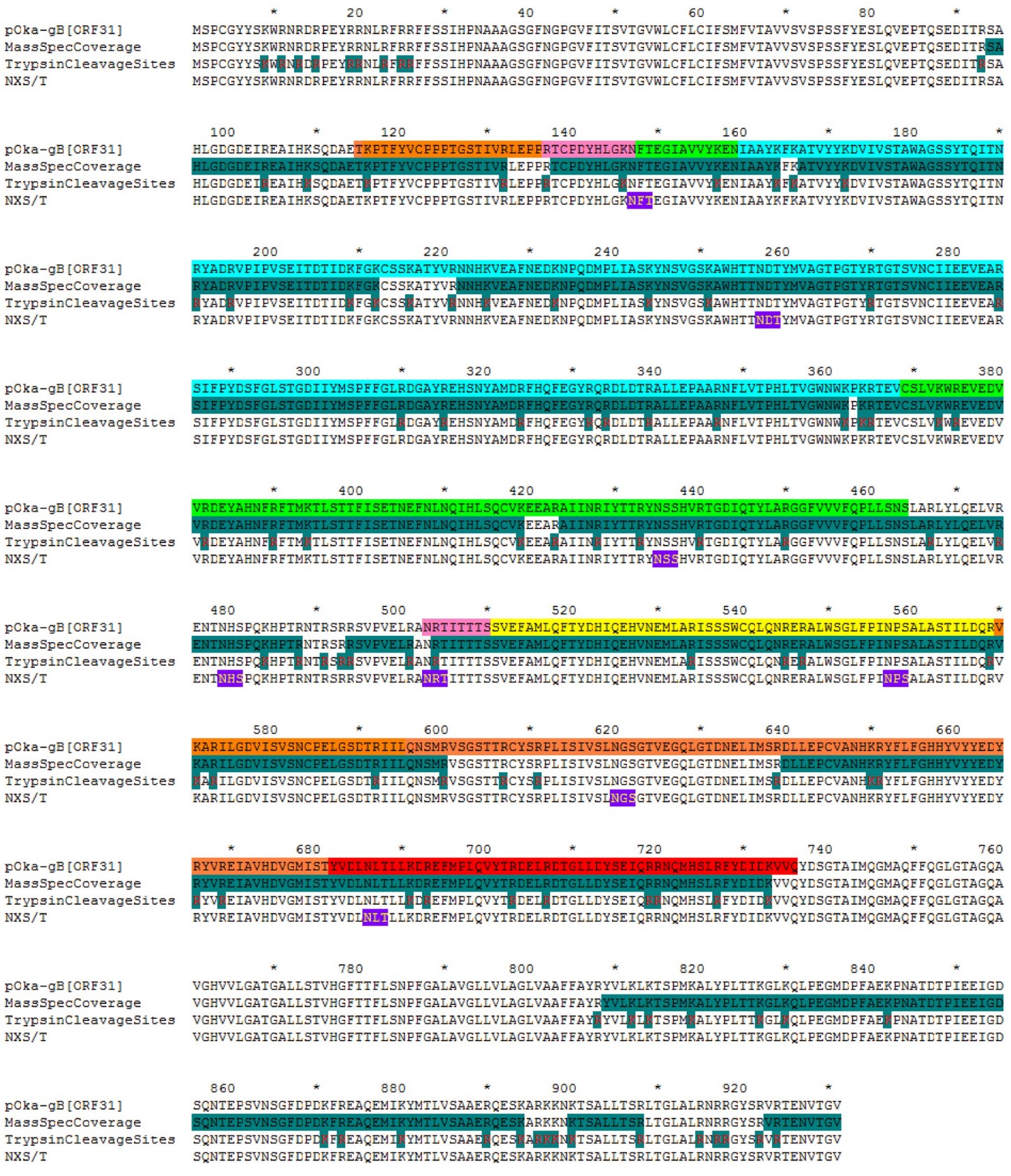

**Fig 6. Peptide coverage from native, full-length VZV gB digested with trypsin and detected by Orbitrap mass spectrometry.** The pOka-gB[ORF31] line shows the domain colors based on the X-ray crystal structure. MassSpecCoverage shades the gB sequence (green, black text) to show the composite peptide coverage generated by trypsin digest and detected by Orbitrap mass spectrometry. TrypsinCleavageSites shades the K/R cleavage sites (green, red text) to show the location of potential trypsin cleavage sites in VZV gB. The NXS/T line shows the location of predicted N-linked glycosylation sites in the gB ectodomain.

superimposable to that of PRV and HSV-1 gB, suggesting that the tryptophan and tyrosine side chains minimally affected the overall conformation around the fusion loop, which was supported by the VZV gB cryo-EM map. The five disulfide bonds in VZV gB were conserved in all of the gB orthologues (**S5 Table**). Of the eight N glycosylation sites predicted for VZV gB, six were confirmed experimentally by the cryo-EM, X-ray crystallography and Orbitrap mass spectrometry data (**S4 Table**). The VZV N-linked glycosylation site at N257 was found only in PRV (N263) whereas glycosylation at VZV gB N620 was conserved for PRV (N636), HCMV (N586) and EBV (N563) but not HSV-1. These differences in reported glycosylation among the gB orthologues might be due to methods of protein production, which were either baculovirus derived or from transiently transfected HEK293 cells for the EBV, HSV, HCMV and PRV structures [2–5,34].

Importantly, the relationship of the VZV gB crown region, DIV, to the remainder of the molecule was similar to HSV-1, PRV and HCMV gB, but differed from the rotation characteristic of EBV gB (**S3 Movie**). A common feature of DIV was the flexibility of the gB N-terminus. Although the gB N-terminus was determined to be flexible for VZV, comparisons of this region to the gB orthologues from PRV and HSV suggested that secondary structure elements in this region might exist to facilitate protein-protein interactions (**Fig 8**). The PRV gB N-terminus formed a β-strand ([115]AVR[117]; VZV aa [108]HKS[110]) and the gB N-terminus was discernable up to aa102 in the HSV-1 (VZV aa108) structure resolved at low pH. In contrast, the HSV-1 gB structure resolved at neutral pH was more similar to VZV. Molecular modelling of VZV gB predicted that residues K109, S110, Q111 and D112 ([109]KSQD[112]) folded into a small α-helix (**Fig 9**). Notably, I53 of mAb 93k VL chain interacts with T115 and K116 of the VZV gB N-terminus (**Fig 10A**). The propensity for this region of the gB N-terminus to form a stabilizing secondary structure in HSV and PRV in addition to the predicted α-helix for VZV and mAb 93k binding suggested that the gB N-terminus might be important for gB fusion function.

## The flexible N-terminus contributes to the fusion function of VZV gB

Based on the molecular modelling, mutagenesis of the unresolved region of the N-terminus was used to establish whether the [109]KSQD[112] residues played a functional role in membrane fusion (**Fig 10B–10F and S2 Spreadsheet**). When the effects of individual alanine substitutions of [109]KSQD[112] residues were compared to WT gB in a stable reporter fusion assay (SRFA), gB/gH-gL mediated fusion was reduced to 42% by K109A, 58% by S110A and to <10% by Q111A and D112A (**Fig 10B**). A quadruple substitution, [109]AAAA[112], abolished fusion. Thus, [109]KSQD[112] residues in the gB N-terminus were important for fusion function attributed to the potential of these residues to form an α-helix. In contrast to K109A, fusion was not severely impaired by a K109R substitution, suggesting that charge at this residue might contribute to fusion function. Cell surface levels of the gB mutants were similar to WT gB when tested with mAbs 93k and SG2 using flow cytometry of transfected cells (**Fig 10B and S1 Fig**).

To establish whether the role of the [109]KSQD[112] residues in gB fusion, as shown in the SRFA, was important for VZV replication and spread in MeWo cells, the [109]KSQD[112] substitutions were incorporated into the VZV genome using the pOka-TK-GFP BAC [35]. Infectious

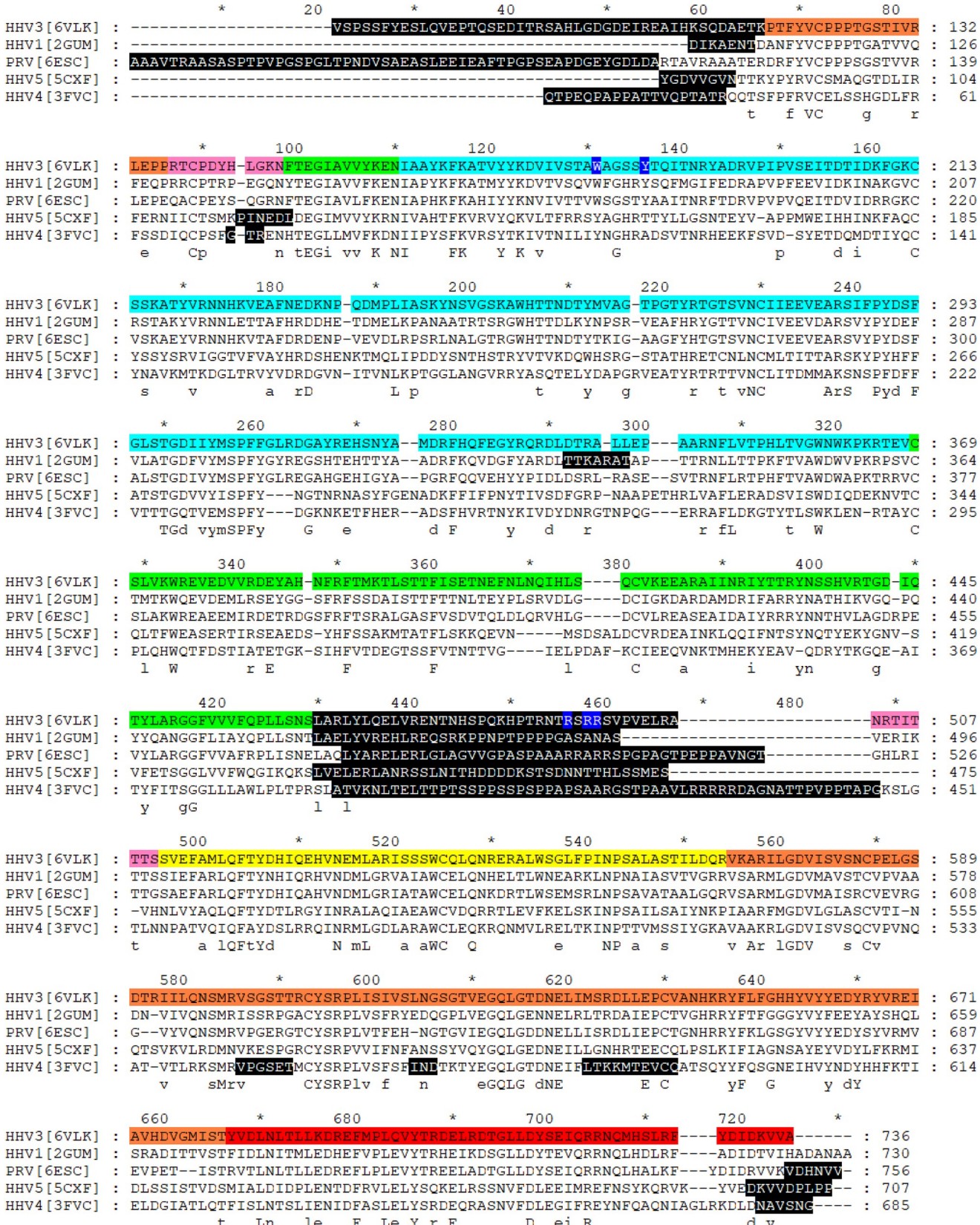

**Fig 7. Structure based amino acid alignments of herpesvirus gB orthologues.** The amino acid sequences given are the partial ones used to express the gB homologs for crystallography studies and are numbered according to the position in the full-length proteins. HSV (HHV1[2GUM] [5]), PRV ([6ESC] [34]), HCMV (HHV5[5CXF] [3]) and EBV (HHV4[3FVC] [2]) compared to VZV (HHV3[6VLK]). The VZV gB domains are colored as per the crystallography DI (cyan), DII (green), DIII (yellow), DIV (orange), DV (red) and linker regions (hot pink). Residues colored with white text and a blue background in the VZV gB amino acid sequence were subsituted for glycine (G) in the transient expression constructs used for X-ray crystallography. Residues colored with white text and a black background were not resolved in the X-ray crystallography data for each of the herpesvirus gB orthologues. The consensus sequence below the alignment depicts conserved (upper case) and partially conserved (lower case) residues in the gB amino acids sequences. The sequences are numbered according to the complete gB for each herpesvirus.

VZV was recovered from cells transfected with the single mutant BACs S110A, Q111A and D112A (**Fig 10C**). Immediate early (IE) 62 and GFP positive cells were seen in transfected MeWo cell monolayers at 72 hours post infection (hpi) indicative of cell-to-cell spread of the VZV mutants. Plaque sizes of S110A, Q111A, D112A mutants were unaltered compared to the BAC expressing WT gB, reflecting the canonical cell-cell spread of VZV in MeWo cell monolayers (**Fig 10C and 10D**). In contrast, the $^{109}$AAAA$^{112}$ mutant had very limited spread in BAC transfected MeWo cells, infectious virus was difficult to recover and plaque sizes were significantly smaller compared to the BAC with WT gB. The K109A and K109R mutations prevented VZV infection, suggesting that altering K109 was the major factor in limiting the propagation of the $^{109}$AAAA$^{112}$ mutant. Of note, the Q111A and D112A mutant BACs produced infectious VZV even though cell fusion was <10% of gB WT in the gB/gH-gL SRFA, indicating that even limited fusion function of the gB N-terminus was sufficient to sustain VZV replication *in vitro*. The eventual recovery of infectious VZV from the $^{109}$AAAA$^{112}$ mutant BAC transfections might have been due to the emergence of a compensatory mutation where a G→A transition occurred resulting in a G452E substitution in gB DII (**S2 Fig**). The differential effect of the K109 substitutions, which inactivated VZV, compared to the fusion defective substitutions of Q111A and D112A implies that gB/gH-gL mediated fusion in the absence of other VZV proteins is less tolerant to gB N-terminus manipulation.

The epitopes of SG2 and 93k mAbs were not affected by any of the $^{109}$KSQD$^{112}$ substitutions as determined by immunoprecipitation of the gB mutants from transfected cells or by western blot under denaturing conditions (**Fig 10E and 10F**). The preservation of binding of the neutralizing 93k mAb indicated that impaired fusion associated with the $^{109}$KSQD$^{112}$ mutations reflected an independent contribution of the extreme N-terminus to gB fusion rather than an effect of these mutations on the DIV structure recognized by the 93k mAb that are critical for the initiation of gB fusion. The gH-gL heterodimer is critical for the fusion function of gB [9]. The VZV gB/gH-gL fusion complex can be immunoprecipitated using mAb 93k or an anti-V5 mAb to bind gH-V5 [33]. To demonstrate the specificity of this interaction, CHO cells were co-transfecting with the gB/gH-gL vectors along with the pBud-gE/gI vector, which simultaneously expresses VZV gE and gI [36]. Although gE was abundantly expressed upon transfection, it did not co-immunoprecipitate with the gB/gH-gL complex (**S3 Fig**). All of the gB N-terminal mutants retained the capacity to form a complex with gH-gL, eliminating the role for $^{109}$KSQD$^{112}$ in this interaction, which is a prerequisite for membrane fusion (**Fig 10F**).

## The neutralizing epitope of mAb 93k is accessible on the prefusion form of VZV gB

Recently, a subnanometer (9.0Å) cryo-EM map of HSV-1 gB has been determined [37]. This was a considerable improvement on previous studies of membrane bound gB that achieved 21Å and 23Å resolution, which led to divergent interpretations about the arrangement of domains I-V in the gB structure [38,39]. The 9.0Å resolution cryo-EM map was sufficient to

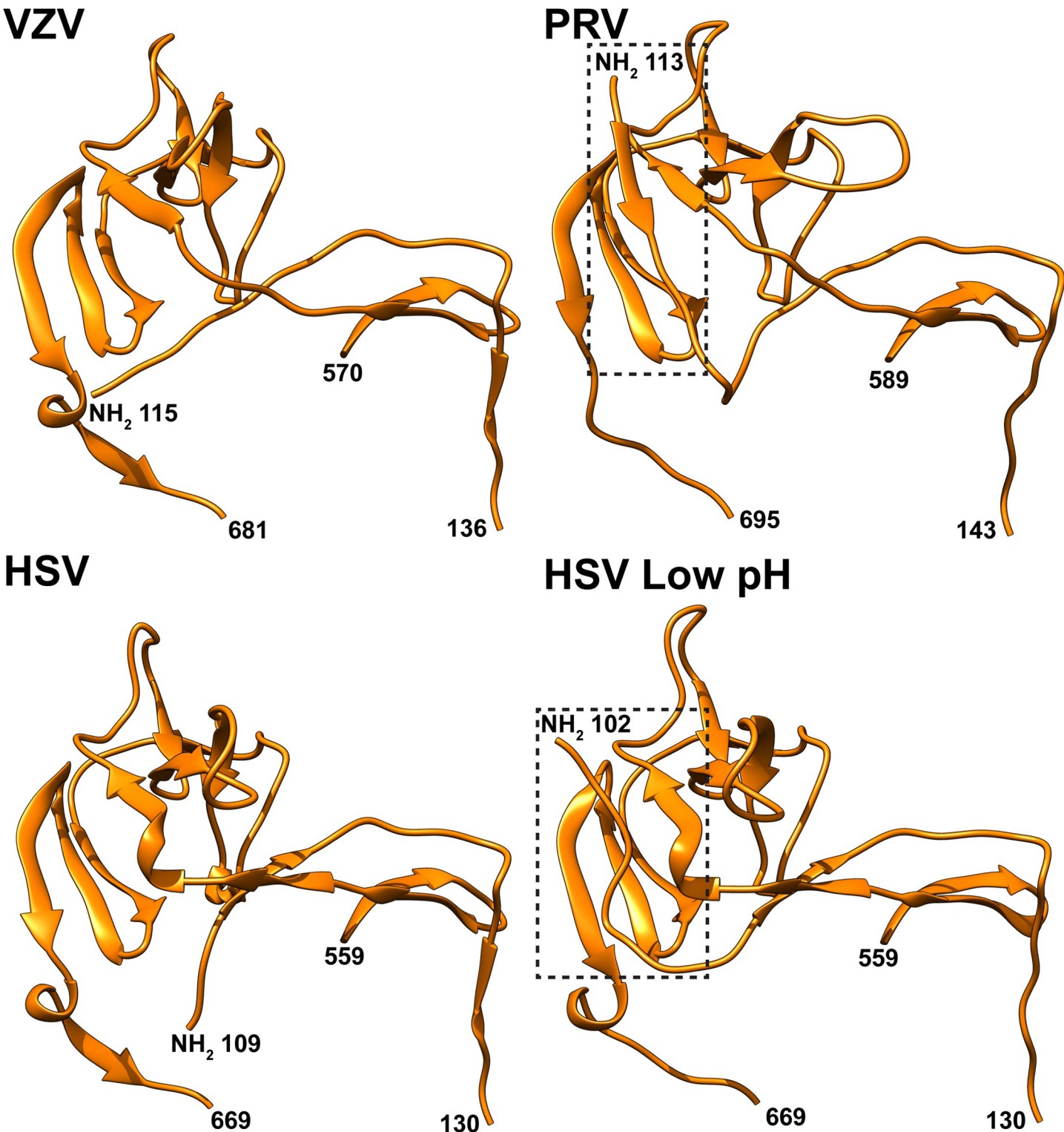

**Fig 8. The N-termini of PRV and low pH HSV gB have an ordered structured compared to those of VZV and neutral pH HSV.** The gB DIV structures for VZV (PDB), PRV (6ESC[34]), HSV (2GUM[5]) and low pH HSV (3NWF[87]) gB are represented in ribbon format. Residues in the gB structure for the polypeptide chain are indicated. The ordered N-terminal regions of PRV and low pH HSV gB are highlighted by the dotted boxes.

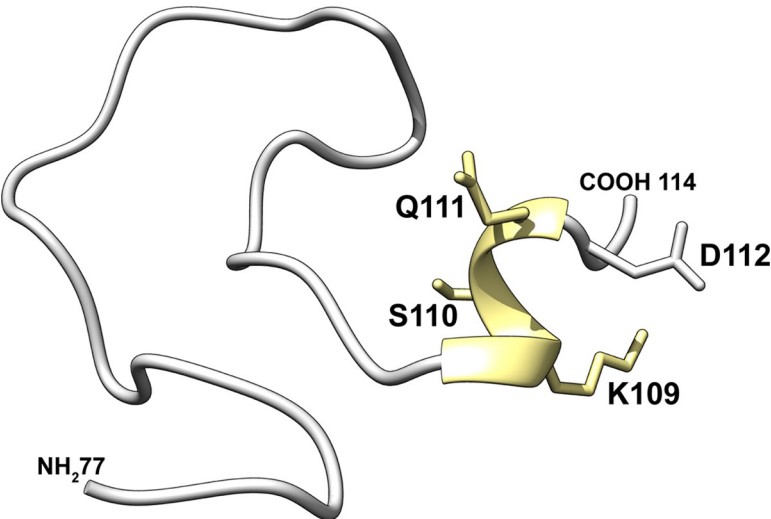

**Fig 9. Homology model of the VZV gB N-terminus.** Residues 77–114 of VZV gB were subject to homology modelling using RaptorX [88]. The predicted α-helix is colored yellow.

assign the location of individual domains of gB into the map density and generate a model of the prefusion form of HSV-1 gB. Importantly, the location of gB DIV was unambiguously assigned. To evaluate the location of the mAb 93k and SG2 epitopes on the prefusion form of VZV gB, the prefusion model of HSV gB was used to generate a homology model of VZV gB in a prefusion conformation (**Fig 11A and S4 Movie**). As expected, the domain organization and orientation of the VZV gB domains were similar to HSV-1 gB; DI and DII formed a tripod around DIV, which was inverted toward the membrane compared to the postfusion form of gB. DIII was exposed at the top of the trimeric molecule and DV was modelled to fill the space in the crown of DIV. A lipid membrane was also modelled using a lipid composition that was previously determined for HSV virions produced by infected Vero cells (**Fig 11B and S4 Movie**). Critically, the neutralizing epitope of mAb 93k was exposed on the surface of DIV for the predicted prefusion form of VZV gB (**Fig 11C and S4 Movie**). Modelling of the 93k Fabs bound to the prefusion form clearly showed that interactions between the N-terminal residues and the two beta strands β23 and β30 were possible. Moreover, the non-neutralizing epitope of mAb SG2 was inaccessible on the prefusion form of VZV gB. The angle that mAb SG2 binds to gB DIV is not compatible with the presence of the lipid bilayer; mAb SG2 would be in conflict with the lipid headgroups of the outer leaflet. These predictions fully explain why mAb 93k can neutralize VZV by inhibiting membrane fusion whereas SG2 cannot. The prefusion model of VZV gB also corroborates the role of the N-terminus in fusion as this domain remains in close proximity to β23 and β30 in DIV and is accessible for interactions with the extra-virion environment.

## Discussion

This comparative structural study of gB epitopes targeted by the neutralizing mAb, 93k, and the non-neutralizing mAb, SG2, identifies a role for the gB N-terminus in fusion function. Specifically, N-terminal residues [109]KSQD[112] of the VZV gB DIV were positioned in close proximity to the footprint of mAb 93k but not mAb SG2, defining a more extended functional region on gB. Mutagenesis of [109]KSQD[112] established the functional importance of these N-terminal residues for membrane fusion in the gB/gH-gL fusion assay and in cells infected with

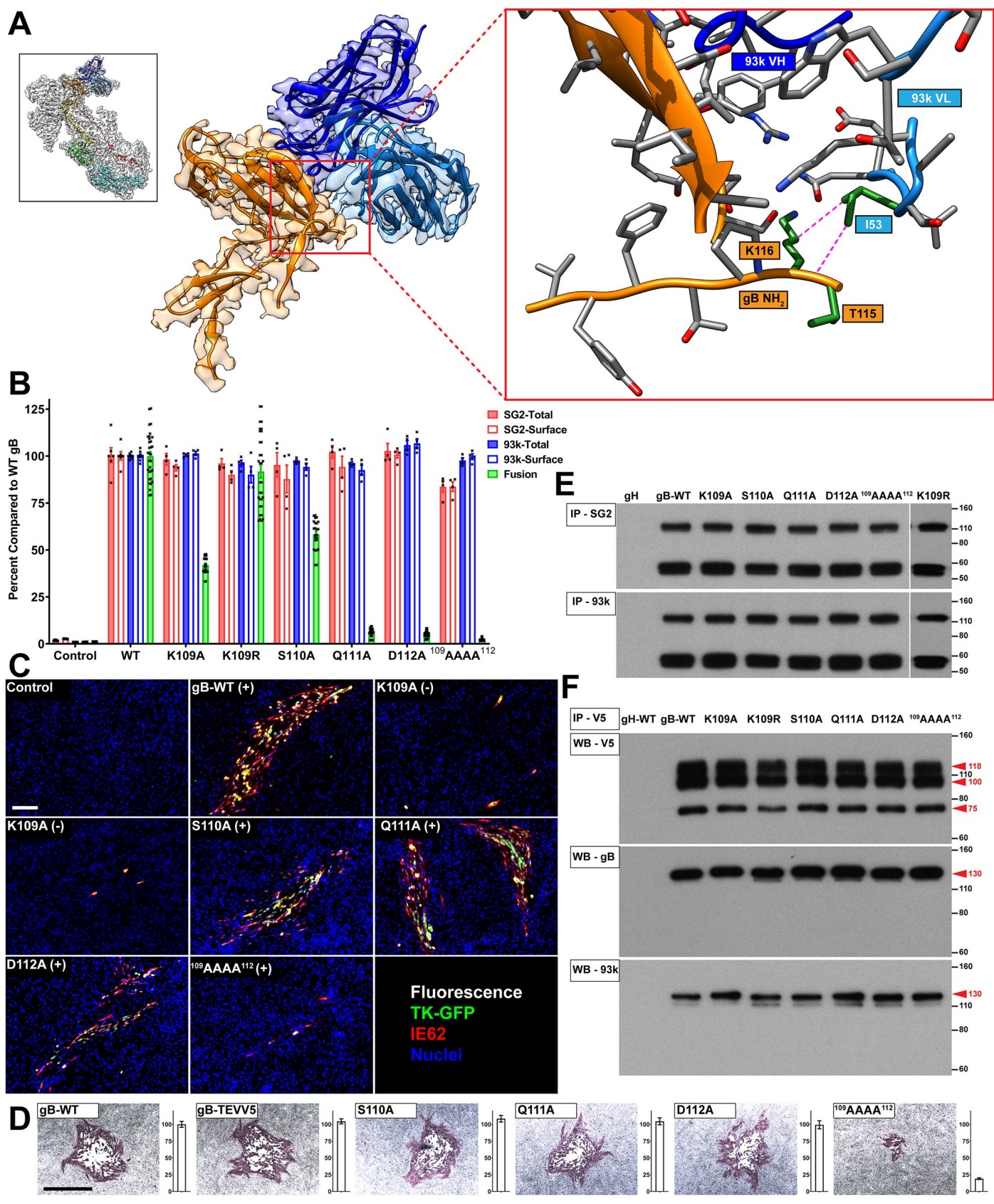

**Fig 10. The gB N-terminus is critical for cell-cell fusion and VZV replication.** A–Near atomic structure of the gB N-terminus within the footprint of mAb 93k [33]. The orientation of the complete gB-93k map is shown in the small box in the top left-hand corner colored as for Fig 1E of [33]. A portion of the cryo-EM map for gB domain IV chain A (Orange) and the bound 93k Fab (Blue) are shown. The red box on the right is a magnified view of the location of the N-terminus with the amino acid side chains (grey) shown for VZV gB and 93k VH and VL chains are shown. Amino acid side chains of the gB N-terminus that interact with 93k VL chain are colored green and interacting atoms connected with a dotted magenta line. The structure for the gB N-terminus is not modelled beyond T115 due to the lack of structure identified for both the cryo-EM map and X-ray crystallography data. B–Quantification of total and cell surface gB DIV mutants produced by transfected CHOs and their capacity for cell-cell fusion measured by the SRFA. All values are normalized as a percentage to WT gB. Bar charts represent n = 4 samples for total and cell surface gB detected using mAbs SG2 and 93k, and at least n = 24 samples for fusion examined over 2 independent experiments. Error bars represent ±SEM. C–Immunofluorescence of MeWo cells at 72 hours post transfection with pOka-BACs with gB mutations. Melanoma cells were transfected with pOka-TK-GFP BACs carrying alanine substitution at K109A, K109R, S110A, Q111A, D112A and [109]AAAA[112]. The (+) or (-) indicate whether virus was recovered or not, respectively, from the transfections. Immunofluorescence staining was performed for IE62 as a marker for early infection because the TK-GFP is a late protein product during VZV replication. Scale bar (white) 100μm. D–Immunohistochemistry staining of plaques and their sizes for the pOka-TK-GFP gB, pOka-TK-GFP gB-STEVV5 and N-terminal mutants K109A, K109R, S110A, Q111A, D112A and [109]AAAA[112]. Scale bar (black) 1mm. Bar charts represent n = 40 plaques measured over two independent experiments. All values were normalized to WT VZV. E–Immunoprecipitation (IP) of the VZV gB N-terminal mutants from transfected CHO cells using anti-gB mAbs SG2 and 93k, and western blot with anti-gB Ab 746–868. The gH lane is a control where CHO cells were transfected with gH-WT. F–Reducing SDS PAGE and western blot of gB co-immunoprecipitated with gH-V5 from CHO cells transfected with the N-terminal mutants, gH-V5 and gL. The first gB-WT lane is a control lane using gH-WT. Western blots were performed using mAb to V5 (Top; WB-V5), anti-gB Ab 746–868 (middle; WB-gB) and a mAb 93k (bottom; WB - 93k). E and F–Numbers to the right of the blots are molecular weight standards (kDa). The red arrows and corresponding numbers indicate molecular weights for gH (118, 100 and 75 kDa; V5 tagging of gH enables the detection of different maturation states of the glycoprotein [89]) and gB (130kDa; only the mature uncleaved form of VZV gB interacts with VZV gH).

VZV gB mutants. While the N-terminal portion of gB associated with DIV formed a helix in both PRV and HSV-1 gB [34,40], such a helical feature in the VZV N-terminus was not discernable in either the X-ray crystallography structure (2.4Å) or the cryo-EM structure (3.9Å) of natively expressed gB both in the absence of bound antibody or the previously reported cryo-EM structure (2.8Å) of the gB-93k interface [33]. In a previous study of HSV-1 gB, linker insertion mutagenesis of the gB N-terminus targeted two regions with differential effects; insertions at K70, K76, P80 and P81 did not affect gB function but insertions at P118 and P119 disrupted gB synthesis and abolished fusion [41]. Insertions at P118 and P119 were located within the region that co-folds with the C-terminal portion of DIV. In contrast, the VZV gB [109]KSQD[112] region does not co-fold with DIV but was determined to be flexible under the conditions used to resolve the gB structure in this and our previous study [33]. Furthermore, each of the mutants that targeted the [109]KSQD[112] region were synthesized and folded correctly as demonstrated by immunoprecipitation with mAbs 93k and SG2, and canonical cell surface levels of gB. This finding indicates that the VZV gB N-terminus (aa71-114) has a role in fusion function independent of interactions with other domains in the gB structure or other potential roles in gB folding.

Importantly, defining the role of the VZV gB N-terminus in fusion function is of significance because the N-terminus of gB homologues has been identified as a site of vulnerability that is targeted by the humoral immune response [33]. For HSV, neutralizing antibody binding sites have been mapped to the gB structure, which have been termed functional regions (FR) 1 to 4 [42]. The gB DI and DV are bound by FR1 mAbs whereas FR2 and FR3 are bound by mAbs to DII and DIV respectively. FR4 (HSV-1 gB residues 31 to 86) corresponds to the N terminus of gB [43–46]. Similar principles have been used for HCMV gB to describe five antigenic domains (AD) AD-1 to 5 [4,47–49]. DI contains AD-5, DII contains AD-4, and the C-terminus of full-length gB after the transmembrane domain contains AD-3. Unsurprisingly, neutralizing antibodies have not been identified for AD3. Relevant to this study, HCMV gB DIV contains AD-1, which is an immunodominant region for neutralizing antibodies [50]. Of interest, AD-2, which is found within the first 80 N-terminal residues of HCMV gB DIV, is a site of neutralization but only about 50% of human sera from HCMV-infected donors have antibodies against this determinant [49]. Although AD-2 is located at the gB N-terminus, it consists of at least two distinct sites between aa 50 and 77 of gB [48,51–53]. Our findings with VZV gB mAb 93k together with the characterization of HSV-1 and HCMV neutralizing

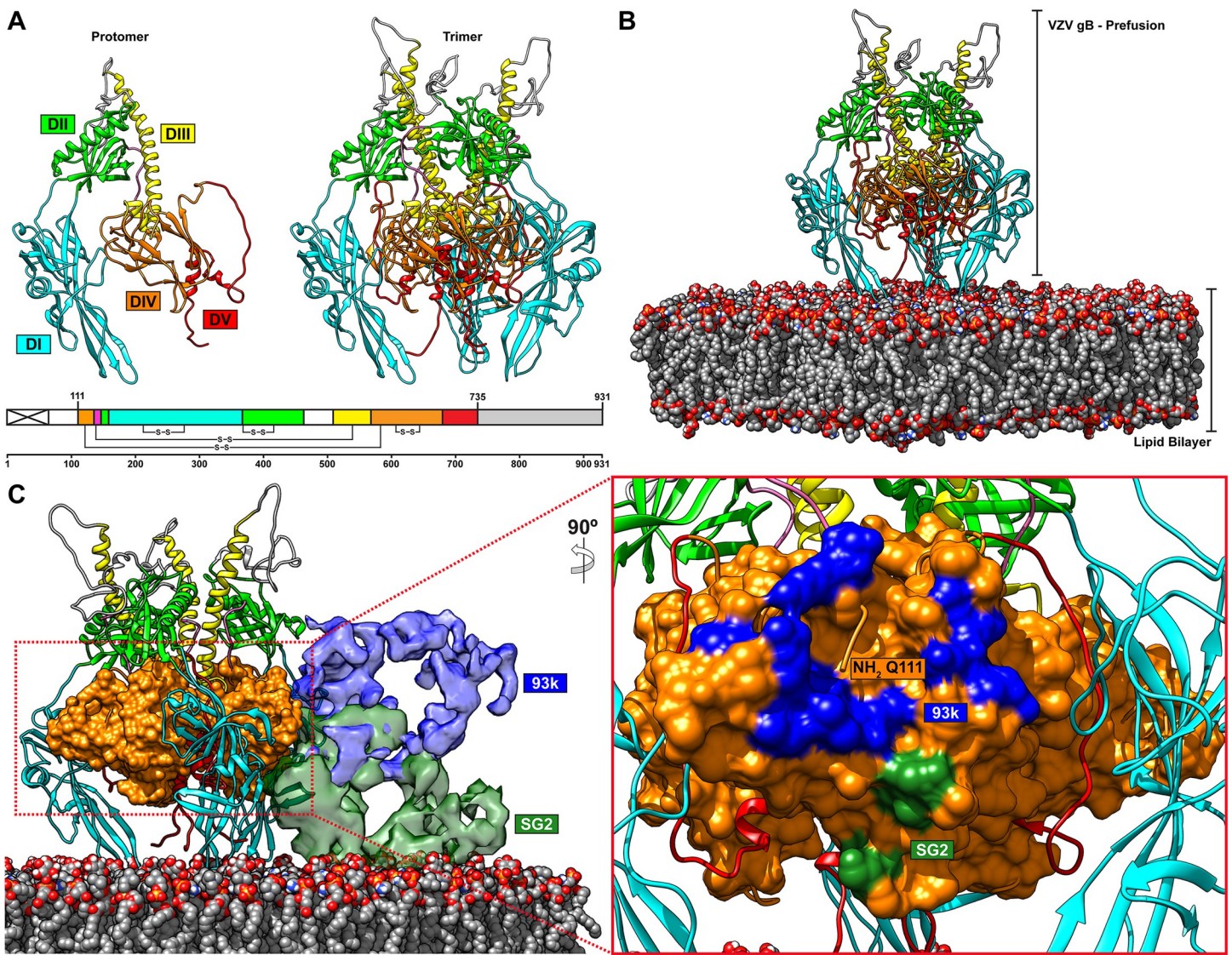

**Fig 11. The neutralizing epitope of mAb 93k is accessible on the prefusion form of VZV gB.** A–A homology model of VZV gB in a prefusion conformation based on the 9.0Å cryo-EM structure of HSV gB [37]. The left-hand panel shows a single VZV gB protomer and the right-hand panel is the complete gB trimer in the prefusion conformation. The linear structure of VZV gB below the two panels depicts the colors used for the homology model; the signal sequence (crossed white box), DI (cyan), DII (green), DIII (yellow), DIV (orange), DV (red), linker region (hot pink) and CTD (grey box). B and C–Snapshots taken from **S4 Movie**. B–The prefusion conformation of VZV gB modelled in conext with a lipid bilayer composed of lipids SM (3.1%), PC (51.2%), PI (11.3%), PS (4.6%), PE (29.8%) found in herpesvirus virions [85]. C–The mAb 93k (blue) and SG2 (green) Fabs bound to the prefusion conformation of VZV gB. Domian IV (orange) of VZV gB is represented in surface. The presence of the lipid bilayer does not prevent accesibility of the 93k epitope but will prevent SG2 from binding to gB in the prefusion conformation. The left-hand panel shows a rotated (90˚) and zoomed in region of gB DIV and the footpring of the 93k and SG2 Fabs as determined in Fig 3.

antibodies that bind to gB DIV, show the importance of this domain for fusion function across the alpha- and beta-herpesviruses [4,47–49,53–55].

In our previous study, residues within VZV gB β23 and β30 were demonstrated to be required for fusion function and the binding of mAb 93k [33]. Furthermore, the footprint of mAb 93k occluded the gB N-terminus and there were interactions between gB T115 and K116 with 93k VLCDR2. These molecular interactions support a potential mechanism of mAb 93k neutralization where VZV gB is prevented from undergoing a conformational change that could be mediated, in part, by the N-terminus. Such a mechanism would partly explain the

disparity in neutralizing activity between mAbs 93k and SG2, with the latter not binding to either the N-terminus or β30, which our previous study and this study demonstrate are essential regions for gB fusion function. Critically, the HSV-1-based prefusion homology model of VZV gB in the context of a lipid bilayer establishes the location of gB DIV and reveals how mAb 93k would bind to this region and the N-terminus to prevent fusion initiation. There have been several studies demonstrating that mAbs can prevent viral fusion proteins from undergoing conformational changes [56–58]. A single study to date provides some evidence that a similar phenomenon is plausible for herpesvirus gB [59]. Compact globular structures were identified by cryo-ET and subtomogram averaging of microvesicles produced by cells co-expressing gB and Fab fragments of an HSV neutralizing mAb, SS55, which binds to gB DI. The compact form of gB resembles the proposed prefusion structure of gB seen on vesicles purified from cells transiently transfected with HSV gB [37,39]. Thus, one hypothesis is that mAb 93k locks VZV gB into a prefusion conformation. An alternative hypothesis is that the binding of mAb 93k to gB prevents an interaction with a cell receptor to trigger conformational change. Both hypotheses are supported by the homology model of prefusion VZV gB in complex with mAb 93k Fabs but the data for the HSV neutralizing mAb SS55 supports the former. In addition to the inability of mAb SG2 to bind the gB N-terminus, mAb SG2 is unlikely to bind the prefusion form of gB based on the VZV gB homology model in the context of a lipid bilayer, which would be necessary to allow interactions with cellular receptors and the conformational change required for membrane fusion.

As we have reported previously, substitutions of the W180 and Y185 residues in one of the predicted gB fusion loops within DI precluded gB/gH-gL cell fusion and prevented VZV infection without altering cell surface levels of gB [11,23]. Similar to other gB studies [2–4], the X-ray crystal structure of the VZV gB ectodomain was derived from a truncated form of the glycoprotein that had W180G and Y185G substitutions in the fusion loop to improve solubility. Comparison of the cryo-EM map derived model of native gB purified from VZV infected cells with the gB crystal structure demonstrated that the W180G and Y185G substitutions did not affect fusion loop topology. In addition, the fusion loop structures were similar to those of gB in the high resolution cryo-EM structure of mAb 93k complexed with VZV gB [33]. These observations suggest that crystal structures of other gB proteins generated with similar loop mutations are likely to reflect the native protein structure [2–5,34]. Furthermore, the VZV gB structures coupled with our previous mutagenesis studies, where the W180 and Y185 mutations inactivate VZV [11,23], provide experimental evidence that amino acid side chains but not the secondary structure of the fusion loop are vital for gB fusion function, as has been proposed previously [34]. Thus, the side chains of amino acids required for fusion function do not alter the peptide topology and are not needed for the switch from a prefusion to a postfusion state because all gB structures, with or without functional fusion loops, transition to a postfusion conformation [2–5,34]. This strongly supports the concept that the gB protein is highly metastable and requires its native lipid environment to maintain a prefusion conformation.

This study highlights the importance of the gB N-terminus and provides structure-function insights into the molecular mechanisms underlying the fusion reaction of herpesviruses. How the gB N-terminus contributes to herpesvirus gB fusion warrants further investigation. The proposed impedance of conformation change by mAb 93k and the lack of functional side chains in the fusion loop set the scene for future studies to characterize the gB structure in the context of infected cells to sample its conformational states. Defining the molecular gymnastics that herpesvirus gB undergoes to achieve entry and cell fusion has the potential to guide the development of novel antiviral therapies.

## Materials and methods

### Reagents, consumables, and resources

All reagents, consumables, and resources, including supplier information where applicable, are provided in S9 Table.

### Cells lines

All cell lines were propagated at 37°C in a humidified atmosphere with 5% $CO_2$. MeWo cells (HTB-65; ATCC) were propagated in minimal essential medium (Corning Cellgro) supplemented with 10% fetal bovine serum (FBS; Invitrogen), nonessential amino acids (100 μM; Corning Cellgro), antibiotics (penicillin, 100 U/ml; streptomycin, 100 μg/ml; Invitrogen) and the antifungal agent amphotericin B (Invitrogen).

### Viruses

The VZV parental Oka strain was originally cloned into a bacterial artificial chromosome (BAC) and designated pPOKA-BAC-DX [60]. The recombinant virus pOka-TK-GFP (pOka-rTK) was generated in a previous study [35]. All recombinant pOka-TK-GFP VZV mutants were derived from the self-excisable BAC, pPOKA-TK-GFP-BAC-DX, as described previously [23]. Briefly, the gB-KAN cassette was digested with BstZ171 and NaeI and the 4,056-bp fragment was gel-purified and used to transform electrocompetent GS1783 *Escherichia coli* carrying the pPOKA-TK-GFP ΔORF31 BAC. After red recombination, the pPOKA-TK-GFP BAC was purified using a large-construct purification kit (Qiagen). BACs were digested with Hind III to verify that spurious recombination had not occurred, and successful incorporation of ORF31 mutations were verified by sequencing the BAC directly. To generate BAC-derived VZV, $10^6$ MeWo cells seeded in 6-well plates (Nunc) 24 hours previously were transfected with 4μg the pPOKA BACs using Lipofectamine 2000 (Invitrogen) following the manufacturer's instructions. Recombinant VZV was typically recovered at 5–10 days post-transfection. All virus stocks, pOka and gB N-terminal mutants, were sequenced to verify that the expected ORF31[gB] sequence was present. Briefly, DNA was extracted from infected cells using proteinase K and phenol/chloroform (Invitrogen). VZV ORF31 was amplified by PCR with KOD Extreme (EMD Millipore) following the manufacturer's instructions using the oligonucleotides [31]F56625-56645/[31]R59697-59717. The PCR products were gel purified and sequenced by Sanger sequencing.

### Anti-VZV gB antibodies

The isolation of human mAb 93k that binds gB and neutralizes VZV, and the preparation of its Fab fragments has been previously described [33]. Mouse mAb SG2 (GeneTex) is a commercially available antibody that binds specifically to VZV gB and used extensively in previous studies [11,15,23,24,33]. Fab fragments were prepared from SG2 using a Pierce Fab Preparation Kit (Thermo Scientific). The rabbit polyclonal antiserum, 746–868, which recognizes the peptide sequence [833]PEGMDPFAEKPNAT[846] in the cytoplasmic region of pOka gB, has been previously described and used extensively in previous studies [11,23,24,33].

### Construction of VZV pOka-TK-GFP-gB-TEVV5

A gB-Kan-TEVV5 shuttle vector was generated in a three-step cloning procedure. First, a gB-Kan-V5 vector was generated by amplifying two fragments from the gB-Kan vector[23] using AccuPrime *Pfx* (Invitrogen) with oligonucleotides gB-AgeI/gB931 and gB-V5/M13R, purified using a QIAquick gel purification kit (QIAGEN) following the manufacturer's

instructions, and ligated into the AgeI and SpeI site of the gB-Kan vector. Secondly, two fragments were amplified from gB-Kan-V5 using AccuPrime *Pfx* with oligonucleotides gB-AgeI/ gB_cterm_Sprotein and gB_link_TEV_link/M13R, gel purified and ligated into the AgeI/SpeI site of gB-Kan to generate the gB-Kan-STEVV5 shuttle vector. The gB-Kan-STEVV5 shuttle vector was used to reconstitute ORF31-STEVV5 into the pOka-TK-GFP-ΔORF31 BAC to generate pPOKA-TK-GFP-gB-STEVV5 and recovery of pOka-TK-GFP-gB-STEVV5 virus was performed as described in the 'Viruses' section.

## Purification of the VZV gB ectodomain for X-ray crystallography

The ectodomain of gB used for crystallography was based on the Oka strain of VZV but codon-modified for optimal expression in *Homo sapiens*. A DNA sequence encoding residues 72–736 that excluded the 71 amino acid signal peptide [61], two glycine substitutions at W180 and Y185 in fusion loop 1, which eliminates fusion, and three glycine substitutions at the arginines in the furin cleavage site at $^{491}$RSRR$^{494}$, was synthesized to improve the stability and solubility of gB. The synthesized gB sequence was sub-cloned into the mammalian expression vector pCMV-III (Novartis AG, Basel, Switzerland) to produce the truncated gB with a 6xHis tag. Transient transfection of HEK293 GnTl$^-$ cells was performed to express gB, which was affinity purified by Ni-NTA affinity chromatography, ion-exchange and gel filtration on a Superdex 200 column (GE healthcare).

## Crystallization of the VZV gB ectodomain, X-ray data-collection and structure determinations

Diffraction quality crystals were obtained for gB from a reservoir solution containing 0.1 M Na acetate pH 4.6, 20% PEG 400, 6% ethylene glycol, using a room temperature vapor diffusion method. Crystals were Frozen in 26% PEG 400 and 6% ethylene glycol. Data sets were collected at 2.4 Å (**S2 Table**). Molecular replacement (MR) using the crystal structure of monomer HSV-1 gB (PDB: 2GUM), yielded a clear solution for gB. There are two copies of the monomer per asymmetric unit. Initial model building was based on NCS-averaged electron density maps. NCS-restrains were applied to the NCS copies throughout the model refinement. Phaser, Phenix and Coot were used for MR, density modification, structure refinement and model building, respectively.

## Purification of native, full-length gB from VZV infected cells

MeWo cells were infected with the pOka-gB-STEVV5 virus and replication was allowed to proceed until cytopathic effect was observed across 90% of the cell monolayer. Infected cells were scrapped into ice cold PBS and pelleted at 424 RCF for 5 minutes. Cells were lysed in glycoprotein extraction buffer (0.1M Tris-base[pH7.2], 0.1M NaCl, 5mM KCL, 1mM CaCl$_2$, 0.5mM MgCl$_2$, 1% sodium deoxycholate, 1% NP40) plus an EDTA free protease inhibitor cocktail (Roche, CA, USA) as described previously [62]. Cell lysates were clarified at 3000 RCF for 10 minutes and 5ml of clarified lysate was incubated with 250μl of anti-V5 agarose beads (Sigma) for 2 hours at room temperature. The beads were washed extensively in PBS + 0.1% Triton and a final wash with PBS then incubated with PBS containing tobacco etch virus protease (55μg/ml) for 20 hours +4˚C. The TEV cleaved gB was eluted from the beads with TBS pH7.4 containing 1mg/ml lauroylsarcosine (Sigma) and 1mg/ml Amphipol 8–35 (Anatrace). Buffer exchange into TBS pH7.4 and 1 mg/ml Amphipol 8–35 using Amicon Ultra-4 centrifugation filters with a 100kDa cutoff (Millipore). The concentration of Amphipol 8–35 was brought up to 35mg/ml and incubated at room temperature for 4 hours then Bio-Beads SM-2 (Bio-Rad) were added and incubated for 16 hours at 4˚C. The purified native, full-length gB

was resolved on either Native PAGE or denaturing SDS PAGE and either stained with Coomassie (Native PAGE) or Gel Code Blue (SDS-PAGE) following the manufacturer's instructions. To determine that the purified protein was native, full-length gB, western blot was performed by transferring proteins to Immobilon-P membranes (Millipore Biosciences, Temecula, CA) and blocked with 5% BSA. The mouse mAb SG2, the human mAb 93k and a previously described rabbit polyclonal antiserum [11], 746–868, which recognizes the peptide sequence $^{833}$PEGMDPFAEKPNAT$^{846}$ in the cytoplasmic region of pOka gB, was used to detect gB. Horse radish peroxidase conjugated antibodies that detect either mouse, human or rabbit IgG (GE Healthcare Bio-Sciences Corp., Piscataway, NJ) were used and HRP activity detected using ECL plus (GE Healthcare Bio-Sciences Corp., Piscataway, NJ). The native, full-length gB was further purified on a Superose-6 column (GE Healthcare Life Sciences) into TBS pH7.4 to remove aggregates.

### Preparation of mAb 93k and SG2 Fab fragments bound to the native, full-length VZV gB

Native, full-length VZV gB was purified from infected cells as described in the 'Purification of native, full-length gB from VZV infected cells' section except the mAb 93k or SG2 Fab fragments were added in molar excess immediately after the TBS pH7.4 and 1 mg/ml Amphipol 8–35 buffer exchange. The native, full-length gB plus Fab fragments were incubated overnight at +4˚C on a rotary mixer. The complexes were concentrated using Amicon Ultra 10kDa filter units following the manufacturer's instructions. The concentration of Amphipol 8–35 was brought up to 35mg/ml and incubated at room temperature for 4 hours then Bio-Beads SM-2 (Bio-Rad) were added and incubated for 16 hours at 4˚C. The purified native, full-length gB-Fab complexes were evaluated by Native PAGE and purified on a Superose-6 column (GE Healthcare Life Sciences) into TBS pH7.4 to remove aggregates.

### Grid freezing

Lacey carbon copper 400 mesh grids with an ultrathin layer of carbon or gold Quantifoil R1.2/1.3 300 mesh grids were used for specimen freezing. EM grids were glow discharged for 25 seconds. To each grid 3µl of purified protein was dispensed and immediately plunge frozen in liquid ethane using a Leica EM GP. Optimum chamber humidity and blotting times were determined empirically for each sample and ranged from 95 to 99% and 1.8 to 2.5 seconds.

### Cryo-EM data collection using the Titan Krios

Micrographs for native, full-length VZV gB were captured on a 300kV Titan Krios (FEI) controlled by SerialEM [63] to automate the data collection procedure using a Gatan imaging filter with a slit width of 20 eV. Movie data (11,283 total stacks) were captured with a Gatan K2 Summit (5 µm/pixel) in counted mode with a dose rate of $\sim$1.335 e$^-$/Å$^2$/s per frame and 200 millisecond exposure time per frame and 12-second total exposure time at a nominal magnification of 130,000x and a pixel size of 1.06 Å/pixel on the specimen. The defocus range was 1.5 to 2.0µm.

### Reconstruction of the cryo-EM maps for the native, full-length VZV gB

The motion correction and damage compensation for all movie-mode data were performed using MotionCor2[64]. CTFFIND4 was used to estimate the contrast transfer function parameters [65]. Initially first 100 micrographs were selected to box out particle images using EMAN2's e2boxer.py [66], followed by Relion's [67,68] 2D classification which generated a set

of 2D class averages. The good 2D class averages were selected as templates to box out the particle images from all micrographs using Relion's auto-picking. EMAN2's e2initialmodel.py [66] or Relion's 3D initial model was utilized to build the initial model. A couple of Relion's 2D classifications were first performed to remove junk, and the good classes were selected from 3D classification to do the final 3D auto-refine, for which the number of particles 176,940 were selected for gB. The C3 symmetry was imposed during the 3D auto-refine of Relion.

## Structure determination, validation and visualization of VZV gB

The quality of the cryo-EM map and model of gB was determined using Fourier shell correlation overall resolution estimate [69]. Structure models were generated and refined using Coot [70] and Phenix [71–73]. ResMap was used to calculate local resolution variation [74]. Newly developed Z and Q scoring tools were applied to calculate feature resolvability [75,76]. All structure images were generated using UCSF Chimera 1.13.1 [77]. Movies were generated using the Animation tool in UCSF Chimera 1.13.1.

## Orbitrap Mass Spectrometry of native, full-length VZV gB

Purified native, full-length VZV gB was resolved on Native PAGE gels and the single protein band that migrated to 690kDa was excised from the gel to be processed for mass spectrometry using an Orbitrap Fusion Mass Spectrometer (Thermo Scientific, San Jose, CA). Gel samples were diced into 1 mm cubes, reduced in 5 mM DDT and alkylated with 10 mM acrylamide. Protein was then digested in gel using the appropriate protease, either Trypsin/LysC (Promega) or AspN (Promega). Resulting peptides were extracted, dried, and reconstituted for injection onto an Acquity M-Class liquid chromatograph (Waters Corporation) for LC-MS. In a typical LC-MS experiment, a flow rate of 450 nL/min was used, where mobile phase A was 0.2% formic acid in water and mobile phase B was 0.2% formic acid in acetonitrile. Peptides were separated using analytical columns prepared in-house; columns were made using pulled and packed fused silica with an I.D. of 100 microns. For packing, 1.8 micron C18 stationary phase beads (Dr. Maisch) were used to an approximate length of 25 cm. Peptides were directly injected onto the column, and a gradient of 2–45% B was used, followed by a high-B wash over 80 minutes. The mass spectrometer was operated in a data-dependent fashion using CID fragmentation for MS/MS spectra generation collected in the ion trap. All .RAW data files were checked using Preview (Protein Metrics) to verify instrument parameters and sample quality prior to full analysis. They were processed using Byonic v2.6.49 and v3.2.0 (Protein Metrics) to identify peptides and infer proteins. Protein sequences were derived using an annotated database containing the NCBI human proteome and the target sequences for VZV. Proteolysis was assumed to be fully specific with up to two missed cleavages. Precursor mass accuracies were held within 10 ppm with fragment ions held within 0.4 Da. Protein false discovery was held at 1%, using standard reverse decoy approaches as described previously [78].

## Quantification of cell surface gB for the N-terminal mutants by flow cytometry

CHO-DSP1 cells ($8 \times 10^5$ cells/well) in 6-well plates were transfected with 5μg WT or mutant pCAGGS-gB expression vectors. Cells were dislodged at 24 hours post transfection using an enzyme-free cell dissociation buffer (Life Technologies, Grand Island, NY), washed with PBS then fixed with 1% paraformaldehyde. The fixed cells were washed with PBS then resuspended in FACS staining buffer (DPBS [Dulbecco's Phosphate-Buffered Saline] (Cellgro, Manassas, VA) with 0.2% IgG-free BSA (Jackson ImmunoResearch, West Grove, PA) and 0.1% NaN3

(Sigma Aldrich, St. Louis, MO)) for cell surface staining with anti-VZV gB mAb SG2-2E6 or 93k. A donkey anti-mouse IgG-Alexa Fluor 555 antibody (SG2-2E6) or goat anti-human IgG-Alexa Fluor 488 antibody (93k) (Life Technologies, Grand Island, NY) was used to detect bound anti-VZV gB mAb. Total gB expression was determined by using the same staining protocol except cells were permeabilized using Cytofix/Cytoperm (BD Biosciences, San Jose, CA) before adding the primary antibody and during the staining procedure. Stained cells were analyzed using a FACSCalibur with CellQuest Pro (BD Biosciences, San Jose, CA). FlowJo (TreeStar, Ashland, OR) was used to determine the quantity of total and cell surface gB on the transfected cells. The quantities for gB mutants were normalized to WT gB, which was set at 100%. Experiments were performed with at least two gB mutant clones, each tested in duplicate.

## VZV stable reporter fusion assay (SRFA)

The stable reporter fusion assay for the VZV glycoproteins gB/gH-gL has been reported previously [22] but was adapted for use with a 96-well plate format. CHO-DSP1 cells seeded at $8x10^5$ per well in 6-well plates 20 hours previously were transfected with 1.6 μg each of pCAGGs-gB, pME18s-gH[TL], and pCDNA-gL plasmids with Lipofectamine 2000 following the manufacturer's instructions. At 6 hours posttransfection, the transfected CHO-DSP1 cells were trypsinized, collected by centrifugation at 424 RCF, and resuspended in 1 ml of medium, of which 250 μl of cells were mixed with 0.75ml of Mel-DSP2 cells at $10^6$ cells/ml. The cells were mixed by inversion and 75μl of the suspension was dispensed to at least triplicate wells of 96-well blacked sided optical bottom plates culture plates (Thermo Scientific). At 40 hours post seeding, 50μl membrane permeable coelenterazine-H (5 μM, Nanolight Technology) substrate for five minutes at room temperature. Fusion was quantified by measuring luminescence using a Synergy H1 Multi-mode reader (Biotek). A minimum of two clones were tested in duplicate experiments.

## Immunofluorescence staining of MeWo cells transfected with pPOKA-TK-GFP BAC mutants

To each well of a 12-well plate (Nunclon Delta Surface; Thermo Scientific) a sterile 18mm coverslip (Fisher Scientific) was placed and 2ml of MeWo cells at $2x10^5$/ml was dispensed and incubated overnight. MeWo cells were transfected with 2ug of pPOKA-TK-GFP BACs carrying gB mutants using Lipofectamine 2000 following the manufacturer's instructions. At 72 hours posttransfection, the media was aspirated, the coverslips washed with PBS and fixed with 4% paraformaldehyde for 10 minutes. Immunofluorescence was performed by blocking the cells with PBS + 10% normal donkey serum (NDS) + 0.1% Triton X-100 then adding a mouse mAb to the immediate early protein IE62 in PBS + 1% NDS + 0.1% Triton X-100. The anti-IE62 mAb was detected with the donkey anti-mouse IgG-Alexa Fluor 555 and nuclei were stained with Hoechst 33342 in PBS + 1% NDS + 0.1% Triton X-100. Coverslips were mounted on glass slides (Selectfrost; Fisher Scientific) using Fluoromount-G (SouthernBiotech) and a minimum of five images were captured for each transfection using a Keyence fluorescence microscope using a 20X objective.

## Quantification of plaque sizes for the VZV gB N-terminal mutants

MeWo cells were seeded at $10^6$ cells/well 24 hours prior to inoculation with 50 pfu of either wild type pOka or gB N-terminal mutants. Each well of the 6-well plate was fixed at 4 days post inoculation with 4% formaldehyde and stained by immunohistochemistry. Images of stained plaques (n = 40) were digitally captured, the stained plaque was outlined, and the area (mm²)

was calculated using ImageJ (National Institute of Mental Health). Statistical analyses were performed using Prism (GraphPad Software).

## Immunoprecipitation of VZV gB N-terminal mutants

CHO-DSP1 cells seeded in 6-well plates were transfected with 5μg/well of pCAGGS-gB vectors carrying the N-terminal mutations using Lipofectamine 2000 following the manufacturer's instructions. At 24 hours post transfection cells were lysed with glycoprotein lysis buffer, the same buffer used for the purification of native, full-length VZV gB, and snap frozen in liquid nitrogen and stored at -20˚C. The SG2 or 93k mAbs were cross-linked to immobilized protein A (Pierce, Rockford, IL) [79]. Each 20μg of mAb was incubated with 30μl protein A beads for 1 hour at room temperature on a rotary mixer. The beads were washed with DPBS then mAbs were cross-linked to the beads with 0.2M sodium borate [pH9.0] and 20 mM DMP for 30 minutes. The cross-linking reaction was quenched with 0.2M NaCl and 0.2M ethanolamine [pH8.0] for 2 hours at room temperature. The cross-linked beads were washed with DPBS. Lysates from the pCAGGs-gB transfected CHO-DSP1 cells were divided equally and incubated overnight at +4˚C with either the SG2 or 93k cross-linked beads. The beads were washed extensively with DPBS + 0.1% Triton X-100 and a final wash of DPBS to remove the Triton X-100. Bound proteins were eluted into sodium dodecyl sulfate (SDS) sample buffer (Bio-Rad) containing 5% 2-mercaptoethanl (Sigma) by incubating the beads at 100˚C for 5 min. Denatured samples were resolved on SDS-polyacrylamide gel electrophoresis precast gels (Bio-Rad, Hercules, CA) and western blot was performed using the 746–868 rabbit poly clonal IgG.

## Immunoprecipitation of VZV gB N-terminal mutants in complex with gH-gL

CHO-DSP1 cells were transfected as described in the previous section with pCAGGS-gB vectors carrying the N-terminal mutations, pME18s-gH[V5] and pCDNA3.1-gL (1.6μg of each vector). To determine the specificity of the V5 immunoprecipitation of the gB/gH-gL complex, CHO-DSP1 cells were transfected pCAGGS-gB, pCDNA3.1-gL, pME18s-gH[WT] or pME18s-gH[V5], and the previously well characterized pBud-gE/gI vector [36] (1.5μg of each vector). At 24 hours post transfection cells were lysed with glycoprotein lysis buffer and snap frozen in liquid nitrogen and stored at -20˚C. The gB/gH-gL complexes were immunoprecipitated with anti-V5 agarose (Sigma). Wash steps, protein elution and SDS-PAGE were performed as outlined in the previous section. Western blots were performed using either mouse anti-V5 tag (Bio-Rad), 746–868 rabbit poly clonal IgG, mAb 93k or mouse anti-gE (Millipore). The anti-gE mAb is very sensitive and can detect low concentrations of gE [36,80–82], critical for demonstrating the specificity of the gB/gH-gL immunoprecipitation.

## Homology model of prefusion VZV gB and localization of mAb 93k and SG2 Fab fragments

SWISS-MODEL (https://swissmodel.expasy.org/) [83] was used to generate the homology model of VZV gB in its prefusion conformation using the 9.0Å cryo-EM structure of HSV-1 prefusion gB (PDB 6Z9M [37]) as a template. A lipid membrane was generated using CHARMM-GUI (http://www.charmm-gui.org/) [84]. The lipid bilayer composition (sphingomyelin (SM), 3.1%; phosphatidylcholine (PC), 51.2%; phosphatidylinositol (PI), 11.3%; phosphatidylserine (PS), 4.6%; phosphatidylethanolamine (PE), 29.8%) was based on the lipids previously determined to be present in herpesvirus virions produced by infection of Vero cells [85]. To localize the epitopes of mAbs 93k and SG2 on the prefusion model of VZV gB, DIV of

gB from the 2.8Å cryo-EM map and model of 93k Fabs bound to VZV gB in its postfusion conformation [33] was aligned to the prefusion homology model of VZV gB. The subnanometer cryo-EM maps of 93k (7.3Å) and SG2 (9.0Å) Fabs bound to VZV gB were then aligned to the 2.8Å gB-93k map. Segger v1.9.5 [86] was used to segment the 93k and SG2 Fab fragments of the subnanometer maps. Structure images and movies were generated using UCSF Chimera 1.13.1 [77].

## Quantification and statistical analysis

**Statistical analysis.** All quantitative data were analyzed with two-way ANOVA to determine statistical significance using Prism (GraphPad Software).

## Supporting information

**S1 Fig. The detection of VZV gB N-terminal mutants on transfected CHO cells by mAbs SG2 and 93k using flow cytometry.** A and B–Histograms for total and surface stained gB for wild type (WT-gB) and the N-terminal mutants K109A, K109R, S110A, Q111A, D112A and $^{109}$AAAA$^{112}$ are presented with fluorescence intensity along the abscissa and frequency along the ordinate. In all of the histograms the negative control (gH) is shaded grey, the positive control (WT-gB) a solid line and each of the mutants a dotted line. Numbers above the gates (|—|) are the percentage of positive events for either surface stained or total gB as determined by either SG2 of 93k staining.
(TIF)

**S2 Fig. Sanger sequencing of VZV gB N-terminal mutant virus stocks.** PCR products generated from ORF31[gB] of virus stocks were sequenced to determine that only the expected mutations were present for each of the VZV gB N-terminal mutants generated from the transfection of MeWo cells with BACs. Electropherograms in the N-terminal (A) and compensatory mutation (B) regions are shown with the codon for each of the alanine substitutions underlined. A–The coding DNA sequence and translated amino acids for gB-WT are provided under each panel with the substituted amino acids highlighted in red. B–A G→A transition occurred in the $^{109}$AAAA$^{112}$ mutant resulting in a G452E substitution in gB DII.
(TIF)

**S3 Fig. The VZV gH-gL heterodimer specifically interacts with VZV gB.** Western blots of lysates or anti-V5 immunoprecipitates (αV5) from CHO cells transfected with plasmids expressing either gB/gH-WT/gL/gE/gI (1) or gB/gH-V5/gL/gE/gI (2). Western blots were performed using the same samples with the anti-gB human mAb 93k (αgB), mouse mAb anti-V5 (αV5), and mouse mAb anti-gE (αgE). Numbers to the right of the blots are molecular weight standards (kDa).
(TIF)

**S1 Table. Cryo-EM data collection parameters for the native, full-length VZV gB (EMDB 22629), and the gB-93k (EMDB 22519) and gB-SG2 (EMDB 22520) complexes.**
(DOCX)

**S2 Table. X-ray data collection and structure refinement for VZV gB (PDB 6VLK).**
(DOCX)

**S3 Table. Amino acid residues and color code for each domain in VZV gB.**
(DOCX)

**S4 Table. N-linked glycosylation sites identified in VZV and herpesvirus gB orthologues.**
(DOCX)

**S5 Table. Conserved cysteine bonds in herpesvirus gB orthologues.**
(DOCX)

**S6 Table. Amino acid residues from X-ray crystallography data for the herpesvirus gB orthologues used to calculate amino acid identities and RMSD.**
(DOCX)

**S7 Table. Amino acid identities of VZV gB derived from structure-based alignments with herpesvirus gB orthologues.**
(DOCX)

**S8 Table. RMSD of VZV gB derived from structure-based alignments with herpesvirus gB orthologues.**
(DOCX)

**S9 Table. Key reagents and resources.**
(DOCX)

**S1 Movie. Subnanometer resolution cryo-EM structures of Fab fragments from either 93k or SG2 in complex with native, full-length VZV gB purified from VZV infected MeWo cells.** This movie provides a comparison of the two subnanometer cryo-EM maps derived for the gB-93k (7.3Å; EMDB 22519; 93k –blue) and the gB-SG2 (9.0Å; EMDB 22520; SG2 –green) complexes. The gB ectodomain is shown in grey and the CTD of gB show in red. The ribbon structure of gB DIV is colored orange.
(MP4)

**S2 Movie. Near atomic resolution structures of the VZV ectodomain derived by cryo-EM and X-ray crystallography in the absence of antibody.** This movie compares the cryo-EM and X-ray crystallography structures of VZV gB. The cryo-EM map of native, full-length VZV gB constrained to C3 symmetry (3.9Å; EMDB 22629) and the model starts with the three pro-tomers highlighted as white, blue and green. The white protomer transitions to colored domains; DI (cyan), DII (green), DIII (yellow), DIV (orange), DV (red) and linker regions (hot pink). A segmentation of the cryo-EM map is performed at fusion loop one to demonstrate the sidechain resolvability of gB W180 and Y185, which were absent in the X-ray crystallography structure of VZV gB (2.4Å; PDB 6VLK). The complete X-ray crystallography struture (grey) is compared to the cryo-EM derived model of VZV gB.
(MP4)

**S3 Movie. The arcitectures of herpesvirus gB orthologues.** This movie compares the VZV gB X-ray crystallography structure (2.4Å; 6VLK) to orthologues from alpha- beta and gammaherpesviruses. The VZV gB domains are colored as per the crystallography structure; DI (cyan), DII (green), DIII (yellow), DIV (orange), DV (red) and linker regions (hot pink). The movie shows a single VZV gB protomer rotating around its Y-axis then compared to each herpesvirus orthologue colored grey; HSV-1 (2.1Å; 2GUM [1]), PRV (2.7Å; 6ESC [2]), HCMV (3.6Å; 5CXF [3]) and EBV (3.2Å; 3FVC [4]).
(MP4)

**S4 Movie. Accessibility of mAb 93k and SG2 epitopes on the prefusion form of VZV gB.** This movie depicts a homology model of the prefusion form of VZV gB and the binding of Fab fragments from mAbs 93k (blue) and SG2 (green) in the context of a lipid bilayer. The

VZV homology model was based on the 9.0Å cryo-EM structure of HSV-1 gB [5]. VZV gB domains are colored cyan (DI), green (DII), yellow (DIII), orange (DIV), red (DV) and hot pink (linker region). The lipid bilayer was modelled using CHARMM-GUI composed of lipids SM (3.1%), PC (51.2%), PI (11.3%), PS (4.6%), PE (29.8%) found in herpesvirus virions [6]. (MP4)

**S1 Spreadsheet. Peptides identified by mass spectrometry of native, full length VZV gB purified from infected MeWo cells.** This spreadsheet provides information relating to the Orbitrap mass spectrometry identification of peptides of the gB N-terminus. (XLSX)

**S2 Spreadsheet. Statistical analyses for cell surface levels, cell fusion and plaque sizes of gB N-terminal mutants.** This spreadsheet provides the details for statistical analyses performed on the plaque sizes of VZV gB N-terminal mutants compared to WT pOka. (XLSX)

**S1 Validation Report. This is the validation report for the 7.3Å cryo-EM map of the VZV gB-93k complex EMD-22519.** (PDF)

**S2 Validation Report. This is the validation report for the 9.0Å cryo-EM map of the VZV gB-SG2 complex EMD-22520.** (PDF)

**S3 Validation Report. This is the validation report for the 3.9Å cryo-EM map and model of VZV gB EMD-22629; PDB-7K1S.** (PDF)

**S4 Validation Report. This is the validation report for the 2.4Å X-ray crystallography structure of VZV gB PDB-6VLK.** (PDF)

## Acknowledgments

We thank the Stanford-SLAC Cryo-EM Facility and the Stanford Bio-X Interdisciplinary Initiatives Program. We thank the Roger Kornberg lab at Stanford University for providing essential reagents and cryo-EM access.

## Author Contributions

**Conceptualization:** Stefan L. Oliver, Yi Xing, David A. Bushnell, Andrea Carfi, Wah Chiu, Ann M. Arvin.

**Data curation:** Stefan L. Oliver, Yi Xing.

**Formal analysis:** Stefan L. Oliver, Yi Xing, Dong-Hua Chen, Soung Hun Roh, Grigore D. Pintilie.

**Funding acquisition:** Wah Chiu, Ann M. Arvin.

**Investigation:** Stefan L. Oliver, Yi Xing, Dong-Hua Chen, Soung Hun Roh, Grigore D. Pintilie, Marvin H. Sommer, Edward Yang.

**Methodology:** Stefan L. Oliver, Yi Xing, Dong-Hua Chen, David A. Bushnell, Marvin H. Sommer, Edward Yang.

**Project administration:** Stefan L. Oliver, Wah Chiu, Ann M. Arvin.

**Resources:** Stefan L. Oliver, Yi Xing, David A. Bushnell, Marvin H. Sommer, Edward Yang, Andrea Carfi, Wah Chiu, Ann M. Arvin.

**Supervision:** Stefan L. Oliver, Wah Chiu, Ann M. Arvin.

**Validation:** Stefan L. Oliver, Yi Xing, Dong-Hua Chen, Soung Hun Roh, Grigore D. Pintilie.

**Visualization:** Stefan L. Oliver, Dong-Hua Chen, Soung Hun Roh, Grigore D. Pintilie.

**Writing – original draft:** Stefan L. Oliver.

**Writing – review & editing:** Yi Xing, Dong-Hua Chen, Soung Hun Roh, Grigore D. Pintilie, David A. Bushnell, Marvin H. Sommer.

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
