## [Decision Letter · Decision Letter 0]

19 Oct 2020

Dear Dr Oliver,

Thank you very much for submitting your manuscript "The N-terminus of varicella-zoster virus glycoprotein B has a functional role in fusion." for consideration at PLOS Pathogens. As with all papers reviewed by the journal, your manuscript was reviewed by members of the editorial board and by several independent reviewers. In light of the reviews (below this email), we would like to invite the resubmission of a significantly-revised version that takes into account the reviewers' comments.

I am writing to you on behalf of the Plos Pathogens’ Editorial Board to thank you for submitting the above referenced manuscript. Your manuscript was reviewed by three experts in the field. I apologize for the tardiness of my letter, I had some trouble getting appropriate reviewers. Whereas two reviewers were very positive and had only relatively minor comments, the third reviewer felt that I should reject your manuscript for publication since the conclusions of your studies are based on binding of your antibody to the post fusion form of gB and not the prefusion form of gB. It seems that to appropriately address this comment you should include an analysis of your structures of DIV bound with the Fabs and considered them in the context of a homology model for prefusion VZV gB using the HSV prefusion form of gB that was recently published in Science. Unfortunately, this data was not available to you when you submitted your manuscript. You may also want to consider the CMV gB structure published in Plos Pathogens in 2018. Once you have done this as well as address the other comments of the reviewers, I am willing to consider your manuscript for publication.

We cannot make any decision about publication until we have seen the revised manuscript and your response to the reviewers' comments. Your revised manuscript is also likely to be sent to reviewers for further evaluation.

Sincerely,

Richard Longnecker

Associate Editor

PLOS Pathogens

Shou-Jiang Gao

Section Editor

PLOS Pathogens

Kasturi Haldar

Editor-in-Chief

PLOS Pathogens

orcid.org/0000-0001-5065-158X

Michael Malim

Editor-in-Chief

PLOS Pathogens

orcid.org/0000-0002-7699-2064

I am writing to you on behalf of the Plos Pathogens’ Editorial Board to thank you for submitting the above referenced manuscript. Your manuscript was reviewed by three experts in the field. I apologize for the tardiness of my letter, I had some trouble getting appropriate reviewers. Whereas two reviewers were very positive and had only relatively minor comments, the third reviewer felt that I should reject your manuscript for publication since the conclusions of your studies are based on binding of your antibody to the post fusion form of gB and not the prefusion form of gB. It seems that to appropriately address this comment you should include an analysis of your structures of DIV bound with the Fabs and considered them in the context of a homology model for prefusion VZV gB using the HSV prefusion form of gB that was recently published in Science. Unfortunately, this data was not available to you when you submitted your manuscript. You may also want to consider the CMV gB structure published in Plos Pathogens in 2018. Once you have done this as well as address the other comments of the reviewers, I am willing to consider your manuscript for publication.

Reviewer's Responses to Questions

**Part I - Summary**

Reviewer #1: This work uses structural and functional evidence to establish the importance of the N-terminus of VZV gB for fusion. The paper reports three new structures: the crystal structure of the VZV gB ectodomain, a cryoEM structure of full-length VZV gB including the cytoplasmic domain, and a cryoEM structure of VZV gB bound to a non-neutralizing MAb SG2. Comparison of gB bound to nAb 93K (examined by cryoEM here and independently in a recent paper) and gB bound to non-nAb SG2 shows that the Ab binding sites are distinct but overlapping. The analysis suggests that the primary site of non-nAb SG2 binding (β25-26) is not critical for gB function, whereas a binding site exclusive to nAb 93K (at the N-terminus) is important for gB function. The structural data leads to a hypothesis that the unresolved flexible gB N-terminus is important for fusion. A predicted helix in the N-terminus (residues 109-112), just upstream from the 93k binding site, is demonstrated to be critical for cell-cell fusion function by site-specific mutagenesis. When mutations at this site are added to virus, virus growth is impaired. The work supports a model that the N-terminus of gB, a target of neutralizing antibodies, is required for the gB conformational change that mediates fusion during virus entry. The study is logically presented and well described. These new structures and the study of function derived from them contribute to the model of herpesvirus entry.

Reviewer #2: This manuscript describes a tour de force characterization of the ectodomain of VZV gB. The authors are commended on their achievement. Their interesting results have led to a few questions that could not have been asked earlier.

1. Introduction, line 73. To avoid confusion by HSV scientists, mention that VZV gB is cleaved. Also tell us the MW of the gB product before cleavage and the MW of the two cleavage products.

2. Results, line 126. The different anti-gB antibody reagents are mentioned at the beginning of the Results. When the reader goes to the Method section, there is only a brief sentence to describe the antibodies. Because of their importance in this study recommend that the authors prepare a separate section in Methods for the 3 Antibodies. Describe the origin of each antibody, even if already described elsewhere. Give IgG subclass if known. Give 1 or 2 references to each antibody, if available.

3. Figure 10C in Results. Almost all VZV strains are highly fusogenic when grown in human melanoma cells. In contrast, VZV induces only small polykaryons in infected fibroblast cells. Did the authors compare syncytial formation and plaque sizes of their mutant gB viruses between melanoma cells and fibroblast cells? They may also want to comment on this differential in the Discussion. Is there a missing cellular factor in some cells that enhances (or inhibits) VZV-induced fusion and that is unknown to us?

4. Figure 10F. This panel is extremely important. The authors state that all VZV gB mutants retained the capacity to form a complex with gH:gL. This finding deserves a few more sentences of description in the Results. This finding also deserves a few more sentences of comparison with HSV gB and gH:gL in the Discussion. In particular, haven’t the authors shown that VZV gB is the true fusogen, thereby confirming earlier results about HSV gB from Eisenberg/Cohen lab? Eisenburg/Cohen proposed that gD assisted gH:gL, and in turn gH:gL assisted gB to become the true fusogen. Is that the conclusion of these authors (Obviously VZV has no gD)? Or is the above statement a misinterpretation of their VZV data? Perhaps add a few comments in Discussion about the D. Atanasiu et al 2016 Journal of Virology article about the cascade of events of the HSV glycoproteins (PMID: 27630245). Also, please add small numbered arrows to each row of bands in each of the 3 smaller panels within panel 10F, to better identify the individual proteins; and expand the legend to panel 10F.

Reviewer #3: This MS reports a cryo-EM structure of the postfusion conformation of VZV gB bound with a non-neutralizing mouse monoclonal Fab, SG2, and compares the result with a previously published structure of the same protein bound with Fab from a neutralizing human mAb, 93k. Both antibodies bind DIV and have partly overlapping epitopes, but the footprint of 93k is adjacent to a proximal segment of the N-terminal region, which is poorly ordered from the N-terminus to residue 115 (the first well-ordered residue in the structure). The proximity focused the authors on a short peptide, residues 109-112. Mutation of those residues to alanine greatly diminished spread in an assay that depends on gB-mediated cell-cell fusion. The authors conclude that the N-terminal segment contributes to the fusion mechanism.

The key problem, of course, is that this is the post-fusion structure, and neutralization by inhibiting fusion depends on binding the prefusion conformer. The prefusion structures of gB from HCMV and HSV have been determined by subtomogram averaging to quite reasonable resolutions, and in those structures DIV is very close to the fusion loops. So the arguments here are irrelevant. At least the authors should have take their structures of DIV bound with the Fabs and considered them in the context of a homology model for prefusion VZV gB. The meaning of the effect of mutation on residues 109-112 is likewise uninterpretable from just the postfusion structure.

**Part II – Major Issues: Key Experiments Required for Acceptance**

Reviewer #1: The authors demonstrate that all of the gB mutants coIP with V5-tagged gH/gL (Fig. 10F). A negative control lacking gH/gL is included, but a specificity control including an unrelated V5-tagged protein (or a gB mutant that does not interact with gH/gL) is not. If the coIP procedure pulls down membrane fragments, gB could be detected in any sample that includes gH/gL, regardless of whether each gB mutant interacts with gHgL. Specificity is especially a concern when using over-expressed constructs of gB and gHgL. If this gB-gH/gL coIP is an established protocol, the authors can include a citation and explanation.

Reviewer #2: No new experiments are requested in my comments in Part I.

Reviewer #3: See summary. This reviewer does not believe that the MS contains enough new information to be worthy of publication.

**Part III – Minor Issues: Editorial and Data Presentation Modifications**

Reviewer #1: The Q111A and D112A mutants were greatly impaired (<10%) in cell-cell fusion but not in virus growth. In contrast, the K109A mutant was modestly impaired (42%) in cell-cell fusion, but the mutation was not tolerated for virus growth. The authors suggest that the poor function of gB-Q111A and gB-D112A in cell-cell fusion when compared to virus growth may indicate that N-terminal gB mutations are tolerated better when other glycoproteins are present (line 276). This explanation does not explain why the K109A phenotype was reversed (i.e. K109A was better tolerated in the cell-cell fusion assay than the virus assay). Can the authors clarify? Even more interestingly, the K109R mutant functioned at near wild-type levels in cell-cell fusion but was not tolerated for virus growth. What other differences between cell-cell fusion and virus-cell fusion could account for that? Are the authors confident in their BAC sequences?

The authors propose that 93k may lock gB in a prefusion state (line 346). 93K must bind to prefusion gB (or an intermediate form) since it is neutralizing, but the structure shows that 93K bound to a postfusion form of gB. Can the authors add to their discussion here, including the idea that 93K can bind to both forms but still neutralize?

93K is able to bind to the AAAA mutant (Fig. 10B). Do the authors think that the AAAA mutant could be trapped in prefusion? I realize no data are available to answer this question, but I am interested in the authors’ perspectives.

Please makes sure that the cryoEM micrographs are at as high of resolution as possible. Seeing an identifiable morphology for the individual particles by eye may not be possible (for example in Fig. 1B), but it would be nice if the reader could.

Figure 2 legend, line 946: The authors could clarify what is meant by the line "The model is not used for gB-93k."

Fig. 4B, line 946: The legend says that western blots of gB probed with 93k and SG2 are shown, but only the PAb western is shown. The legend can be updated.

Fig 5 legend, line 964: The title of the figure refers only to the full-length gB, not the crystal structure in parts F-H.

Fig. 8: The N-terminal region that is more ordered in PRV and HSV Low pH would be clearer if it were colored.

Fig. 10A: In the strongest magnification (on the far right), showing only the sticks may make the figure clearer. Coloring/shading the N-terminus portion of DIV may also help show how 93k binds to the N-terminus.

Fig. 10 legend, liene 1036: The legend refers to + and - signs that are not in the figure.

Reviewer #2: The minor issues that need to be addressed are cited in my comments in Part I.

Reviewer #3: (No Response)

PLOS authors have the option to publish the peer review history of their article (what does this mean?). If published, this will include your full peer review and any attached files.

Reviewer #1: No

Reviewer #2: No

Reviewer #3: No
---

## [Editor Report · Decision Letter 1]

1 Dec 2020

Dear Dr Oliver,

We are pleased to inform you that your manuscript 'The N-terminus of varicella-zoster virus glycoprotein B has a functional role in fusion.' has been provisionally accepted for publication in PLOS Pathogens.

Best regards,

Richard Longnecker

Associate Editor

PLOS Pathogens

Shou-Jiang Gao

Section Editor

PLOS Pathogens

Kasturi Haldar

Editor-in-Chief

PLOS Pathogens

orcid.org/0000-0001-5065-158X

Michael Malim

Editor-in-Chief

PLOS Pathogens

orcid.org/0000-0002-7699-2064
---

## [Editor Report · Acceptance letter]

29 Dec 2020

Dear Dr Oliver,

We are delighted to inform you that your manuscript, "The N-terminus of varicella-zoster virus glycoprotein B has a functional role in fusion.," has been formally accepted for publication in PLOS Pathogens.

Best regards,

Kasturi Haldar

Editor-in-Chief

PLOS Pathogens

orcid.org/0000-0001-5065-158X

Michael Malim

Editor-in-Chief

PLOS Pathogens

orcid.org/0000-0002-7699-2064